# Autistic children sample costly information with increased variability due to inflexible updating
Haoyang Lu [1,2], Hang Zhang [1,3,4,5,6] ✉ & Li Yi [1,3,5,6] ✉

Efficient information sampling is crucial for human inference and decision-making even for young children. It is also closely associated with the core symptoms of autism spectrum disorder (ASD), since both the social interaction difficulties and repetitive behaviors suggest that autistic people may sample information from the environment distinctively. However, the specific ways in which autistic children sample information, especially when facing explicit costs and adapting to environmental changes, remain unclear. Thirty-two autistic and 41 IQ-matched neurotypical children aged five to eight participated in a computerized bead task, where children decided to gather samples sequentially from an unknown target to infer which of the two options was the target. Autistic children showed lower sampling efficiency under costly conditions compared to neurotypical peers, resulting from increased variability in sample numbers across trials, rather than solely systematic sampling bias. Computational models indicated that while both groups shared a similar decision process, autistic children's sampling decisions were less influenced by dynamic changes and more by recently gathered evidence. This led to higher sampling variation and lowered the efficiency of autistic children. These findings offer valuable insights into the cognitive mechanisms underlying fundamental behaviors in autistic children.

In daily decision-making scenarios, such as choosing from a selection of gelatos, people face the challenge of knowing when to stop gathering information and proceed to a decision. This is particularly relevant for children, given their limited prior experience to refer to. Previous studies have found that children and even infants are active information seekers to guide their decisions under uncertainty[1–9]. However, how children modulate their sampling based on the interplay between information gain and costs has not been thoroughly examined. While studies have suggested that young children might be less optimal than older children or adults in trading-off information gain and sampling cost[2,5,10], the intricate cognitive processes underlying the adaptive information sampling, particularly in autistic children, are still elusive (we use identity-first language throughout, respecting community preferences).

Autism spectrum disorder (ASD) is characterized by challenges with social interactions and restricted and repetitive behaviors[11,12], which may be deeply intertwined with how autistic people interact with and sample information from their environment[13]. For example, social interactions necessitate constant information sampling, such as reading facial expressions to infer the mental state of others to keep the conversation going[14]. Indeed, autistic people show atypical gaze patterns in ambiguous or social scenes, sampling the visual environment less efficiently[15–17]. Repetitive behaviors used as a strategy for reducing uncertainty may also contribute to inefficient information sampling due to the prolonged time spent on redundant details[13].

The link between ASD and information sampling was largely unexplored until recent years. Previous studies that directly examine information sampling in autistic people predominantly test autistic adults or adolescents, presenting mixed results. Some suggested autistic adults sample less and decide faster than neurotypical adults[18,19], while others reported more extensive information gathering in similar tasks[20,21]. While these discrepancies might reflect the heterogeneity of autism phenotypes and methodological differences, including task designs, they could suggest that

[1]School of Psychological and Cognitive Sciences and Beijing Key Laboratory of Behavior and Mental Health, Peking University, Beijing, China. [2]Applied Computational Psychiatry Lab, Max Planck UCL Centre for Computational Psychiatry and Ageing Research, Queen Square Institute of Neurology and Mental Health Neuroscience Department, Division of Psychiatry, UCL, London, UK. [3]PKU-IDG/ McGovern Institute for Brain Research, Peking University, Beijing, China. [4]Peking-Tsinghua Center for Life Sciences, Peking University, Beijing, China. [5]Key Laboratory of Machine Perception (Ministry of Education), Peking University, Beijing, China. [6]These authors contributed equally: Hang Zhang, Li Yi. ✉e-mail: hang.zhang@pku.edu.cn; yilipku@pku.edu.cn

ASD may affect one's ability to sample *optimally* depending on situations, instead of merely oversampling versus undersampling. Using an inference task with explicit sampling costs, Lu et al. found that people with higher autistic traits have more varied and less efficient sampling behaviors, due to using a more inflexible strategy of balancing information gain and cost[22]. However, as information sampling may change significantly from childhood to adulthood[2,5,9,10], the information sampling behavior of autistic children remains unclear.

In the present study, we aimed to investigate differences in information sampling between autistic and neurotypical children. We used a child-friendly version of the bead inference task, where children needed to sample information efficiently to balance between information gains and costs with more samples. To perform optimally, children needed to adapt across different conditions varying in evidence strength and information cost to maximize their rewards. Building on previous research[22], we hypothesized that autistic children would exhibit inefficient sampling when facing the complicated trade-off between information gain and cost, which might result from the varied sampling across trials. To further investigate their cognitive mechanisms underlying sampling decision, we employed a series of computational models that examined how children integrated cost-related and evidence-related information. We hypothesized that the inefficiency and higher sampling variability of autistic children could stem from differences in response to dynamic information that constantly varies within or between trials.

## Methods

### Participants

A total of 32 autistic and 41 neurotypical children aged five to eight years old, participated in and completed the study (Table 1; Supplementary Methods 1 for participant exclusion criteria). Gender was reported by parents or guardians. Race, ethnicity, or socioeconomic status data were not collected. The sample size was determined by both resource limits and effect size estimates from a mini meta-analysis of cognitive studies in autistic research, focusing on decision-making, learning, and information sampling (see Supplementary Table 1 for studies and effect sizes included)[23,24]. We conducted a random-effect meta-analysis of 57 effect sizes from 23 studies, finding a mean absolute correlation of $r = 0.406$ (95% CI [0.324, 0.483]). Given the mean effect size with Type I error = 0.05, a total of 44 (or 59) participants would provide 80% (or 90%) statistical power, respectively. Our

final sample of 73 participants met the 80% power threshold even under a conservative assumption (Supplementary Methods 2). Neurotypical children were recruited from regular elementary schools, and autistic children from a special education school and the local community. All the autistic children received a formal diagnosis of autism spectrum condition from pediatricians based on Diagnostic and Statistical Manual—5 criteria[25]. Seventeen autistic children underwent additional assessment using the Autism Diagnostic Observation Schedule (ADOS)[26], five with both the Childhood Autism Rating Scale (CARS)[27] and the Autism Spectrum Quotient—Children's Version (AQ-Child)[28], and 12 only with AQ-Child (Table 1). Full-scale IQ was measured by the Chinese version of the Wechsler scales[29,30]. The study abided by the Declaration of Helsinki and was approved by the Institutional Review Board at Peking University (IRB Protocol #2019-01-02) and obtained oral consents from children and written consents from their parents.

### Stimuli and procedure

The task, based on the "bead task"[31,32], was adapted as an adventure game (Fig. 1a). Each trial presented two islands with varying ratios of dogs to cats. Children were instructed to determine which island they were on by the animals encountered. They could encounter up to 20 animals by pressing a button and then select the corresponding island. They would receive 100 credits for correct responses and zero for incorrect responses. In some conditions, it would cost children 0, 1, or 4 credits per animal. At the end of the experiment, children received their favorite stickers proportional to their accumulated credits, which incentivized strategic sampling for balancing information gain and cost (Fig. 1b).

The experiment contained three blocks for zero-cost, low-cost, and high-cost conditions (Fig. 1a), with the order counterbalanced. Each block nested two mini-blocks with different dog-to-cat ratios (low-evidence: 60%:40% and high-evidence: 80%:20%, respectively) in a random order. Most children completed 96 trials; however, to accommodate the potential distress during the long task, six autistic children completed a shorter version with 48 trials. The experiment typically took 60–75 min, including the time of instruction and practice.

To ensure requisite understanding, the experimenter asked questions about the ratios, costs, and rewards before practice (see Supplementary Methods 3 for the task instruction and comprehension questions). Practice involved 12 trials mirroring the main task, requiring children to be theoretically correct in over 75% of trials to pass; that is, to choose the island having a higher likelihood, irrespective of actual outcomes. Failed attempts led to repeated instruction and practice, with ineligibility after two failures.

Stimuli were presented on a 24-inch monitor via Psychtoolbox in MATLAB 2016b. Children sat 60 cm from the monitor, holding an Xbox controller. Dog and cat icons were matched in style, size, brightness, and contrast to mitigate bias. To ease the difficulty of memorization, ratio and cost information was displayed before each mini-block, and credit and encountered animal were visible throughout trials for reference.

### Statistical analyses

The study was not formally preregistered. Statistical analyses were conducted in R 4.3.2[33,34], using linear mixed models (LMMs) to address imbalanced data. Diagnostic checks confirmed that the data met the assumptions of the linear mixed models. Specifications of (generalized) linear mixed models for behavioral data are set as followed (Fig. 2), using Wilkinson notation (Wilkinson & Rogers, 1973; * represents main effects of predictors and interaction effects between all predictors plus all lower-order terms):

$$\text{LMM1}: \text{Game credits} \sim \text{Group} * \text{Cost} * \text{Ratio} + (\text{Cost} * \text{Ratio}|\text{Participant})$$

$$\text{LMM2}: \text{Judgement [True/False]} \sim \text{Group} * \text{Cost} * \text{Ratio} + (\text{Cost} * \text{Ratio}|\text{Participant})$$

**Table 1 | Children demographic information**

|  | ASD (N = 32)[a] | NT (N = 41)[a] | p value[b] |
|---|---|---|---|
| Gender |  |  | 0.6 |
| Male | 26 (81%) | 30 (73%) |  |
| Female | 6 (19%) | 11 (27%) |  |
| Age | 6.4 (0.8) | 6.5 (0.7) | 0.6 |
| Full-scale IQ | 105.4 (15.7) | 111.4 (14.8) | 0.10 |
| ADOS (raw score) | 13.0 (4.2) [17] |  |  |
| ADOS-SA | 11.0 (3.3) [17] |  |  |
| ADOS-RRB | 2.0 (1.2) [17] |  |  |
| CARS | 34.4 (4.5) [5] |  |  |
| AQ-Child | 82.4 (22.5) [17] | 60.5 (12.1) [34] | 0.001 |

*ASD* autism spectrum disorder, *NT* neurotypical, *ADOS* Autism Diagnostic Observation Schedule, *SA* social affect, *RRB* restricted & repetitive behaviors, *CARS* Childhood Autism Rating Scale, *AQ-Child* Autism Spectrum Quotient—Children's Version.
[a]*n* (%); Mean (SD), [*N*] the number of valid participants when not all participants were measured.
[b]Pearson's Chi-squared test; Welch Two Sample *t* test.

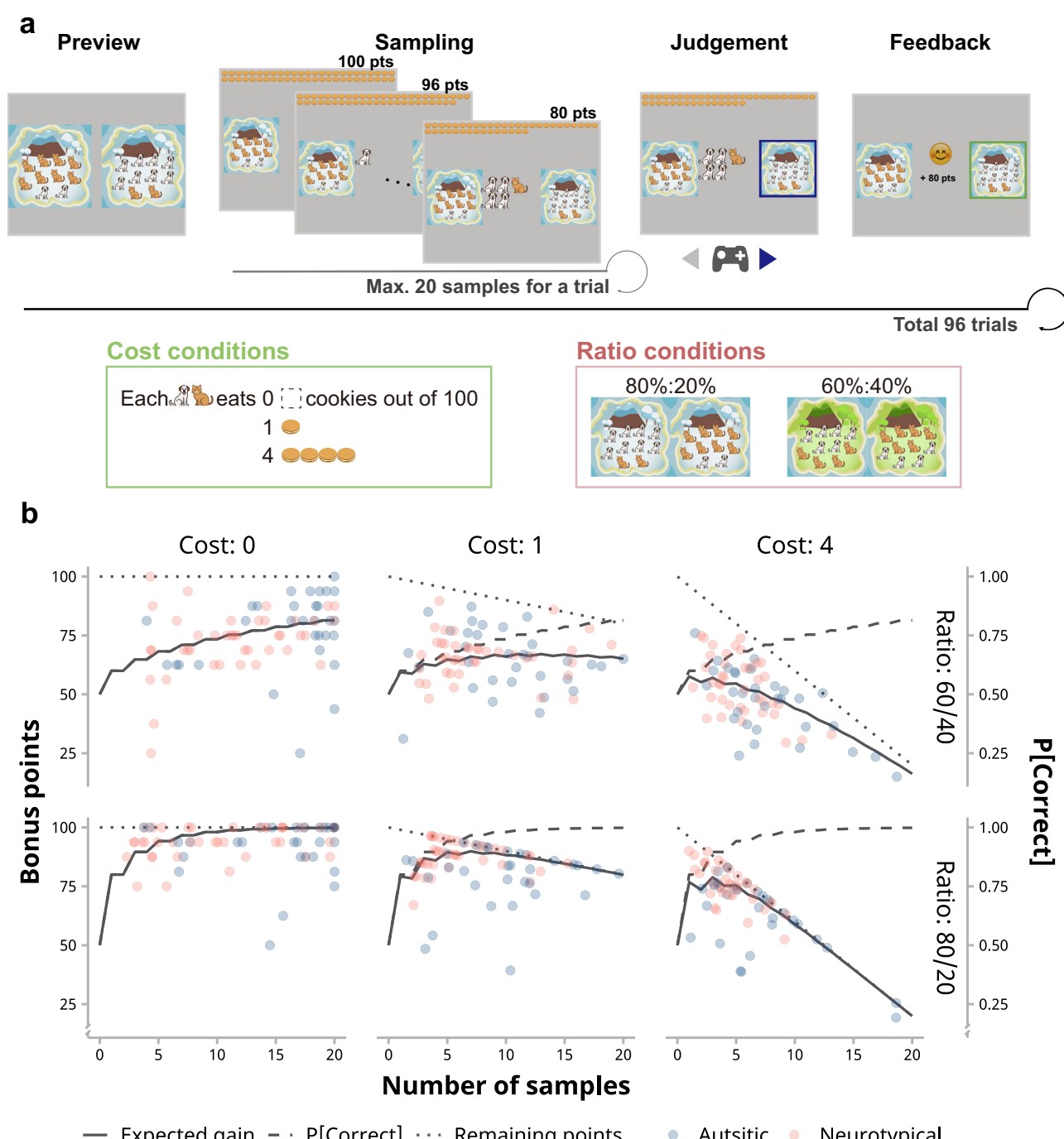

**Fig. 1 | Information sampling "bead" task. a** This child-friendly "bead" task substituted beads with animal images to enhance engagement. Each block began with a preview of the cost and ratio conditions for subsequent trials ("Preview"). During a trial, children might sample up to 20 animals from a predetermined island by pressing a button on the gamepad ("Sampling"). When ready to make a judgment, they could choose by pressing the left or right button on the gamepad ("Judgment"). The task comprised six blocks, each a combination of three cost conditions and two ratio conditions. **b** Children typically faced a trade-off between information gain and cost. The expected probability of a correct judgment (dashed lines) increased, while the remaining bonus points (dotted lines) decreased with the number of samples. Consequently, the expected gain initially increased but then decreased (solid lines, especially noticeable in the "Cost: 1" and "Cost: 4" panels). Each point denotes the average points won by each child across trials ($n = 73$ participants).

$$\text{LMM3} : \text{Efficiency} \sim \text{Group} * \text{Cost} * \text{Ratio} + (\text{Cost} * \text{Ratio}|\text{Participant})$$

$$\text{LMM4} : \text{Signed deviation} \sim \text{Group} * \text{Cost} * \text{Ratio} + (\text{Cost} * \text{Ratio}|\text{Participant})$$

$$\text{LMM5} : \text{Sampling variation} \sim \text{Group} * \text{Cost} * \text{Ratio} + (\text{Cost} + \text{Ratio}|\text{Participant})$$

Efficiency in LMM3 is the same as in Lu et al.[22], which is the expected gain for children's animal sample sizes divided by the maximum expected gain given the sampling cost and ratio condition. It measures approximation to optimal reward-maximizing behavior under different environmental demands. Suppose the animal sample size is $n$, the maximal reward is 100 credits, the unit sample cost is $c$, and the percentage of predominated animals in the preselected isle is $q$, then the expected gain is

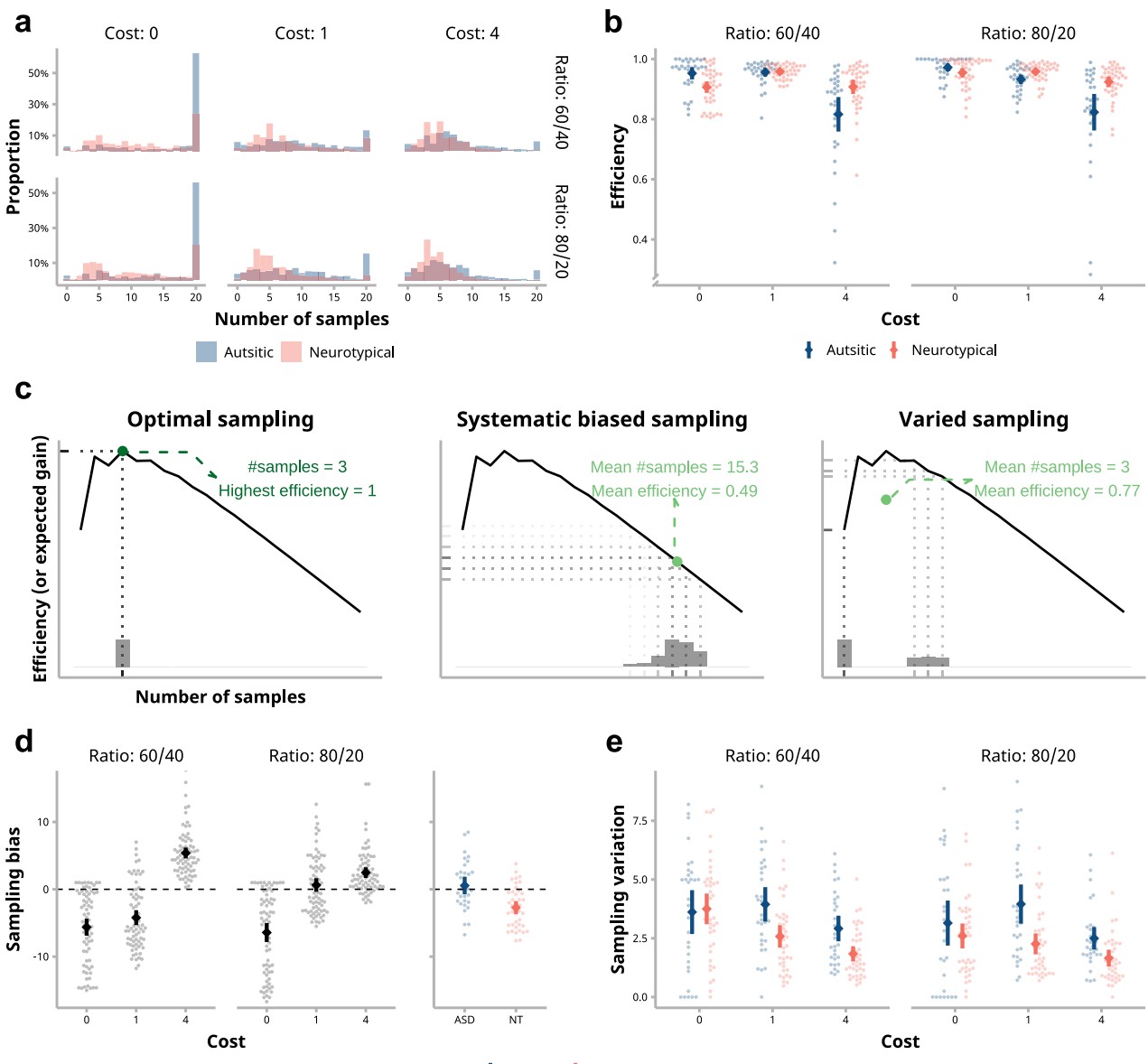

**Fig. 2 | Sampling performance of autistic and neurotypical children.**
**a** Distribution of the average number of samples taken by each child. Children took fewer samples as the sampling cost or sample ratio increased. Autistic children in general sampled more than neurotypical (NT) children. Histograms of both groups are overlaid. **b** Sampling efficiency is the ratio of expected gain from a given number of samples to the maximum expected gain from the optimal sample number. Autistic children exhibited significantly lower sampling efficiency compared to NT children, particularly in high-cost conditions. **c** Illustration of sampling efficiency under different scenarios: optimal sampling (left panel), systematic bias (middle panel),

and large variation in sampling (right panel). Solid lines depict the expected gain as a function of the number of samples, with histograms representing example distributions of samples. **d** Children showed systematic biases in the task. Across conditions, autistic children sampled significantly more than NT children. **e** Sampling variation of autistic children was significantly greater than NT children, particularly under costly conditions. In **b**, **d**, and **e**, large points and error bars denote means and 95% confidence intervals, and each small point represents the mean of each child ($n = 73$ children).

---

$\mathbb{E}\left[\text{Gain}|n, q, c\right] = (100 - nc)p(n|q)$. $p(n|q)$ is the expected probability of correct judgment, defined as follows:

$$
p(n|q) = \begin{cases} \sum_{i=\frac{n+1}{2}}^{n} \binom{n}{i} q^i (1-q)^{n-i} & n = 1, 3, 5, \ldots, 19 \\ \frac{1}{2}\binom{n}{\frac{n}{2}}q^{\frac{n}{2}}(1-q)^{\frac{n}{2}} + \sum_{i=\frac{n}{2}}^{n}q^i(1-q)^{n-i} & n = 2, 4, 6, \ldots, 20 \\ \frac{1}{2} & n = 0 \end{cases}
$$

(1)

Thus, the optimal sample size is the value of $n$ that maximizes $\mathbb{E}[\text{Gain}|n, q, c]$. To investigate the sampling strategy that might result in

inefficient sampling, we further calculate signed sampling deviation (LMM4), which is the signed sample size difference from the optimal sample size, and sampling variation (LMM5), which captures the inter-trial standard deviation of sample sizes in a condition.

In a subset sample of 51 children ($N_{\text{ASD}} = 17$, $N_{\text{NT}} = 34$) who completed the AQ-Child questionnaire, we conducted a series of exploratory, post-hoc analyses to investigate relationships between autistic traits and sampling behaviors, which replaced the group term with the standardized AQ total scores or all the subscale scores (but without the interaction terms between subscales). Given the size of this subsample, we stressed that the results should be interpreted with cautions.

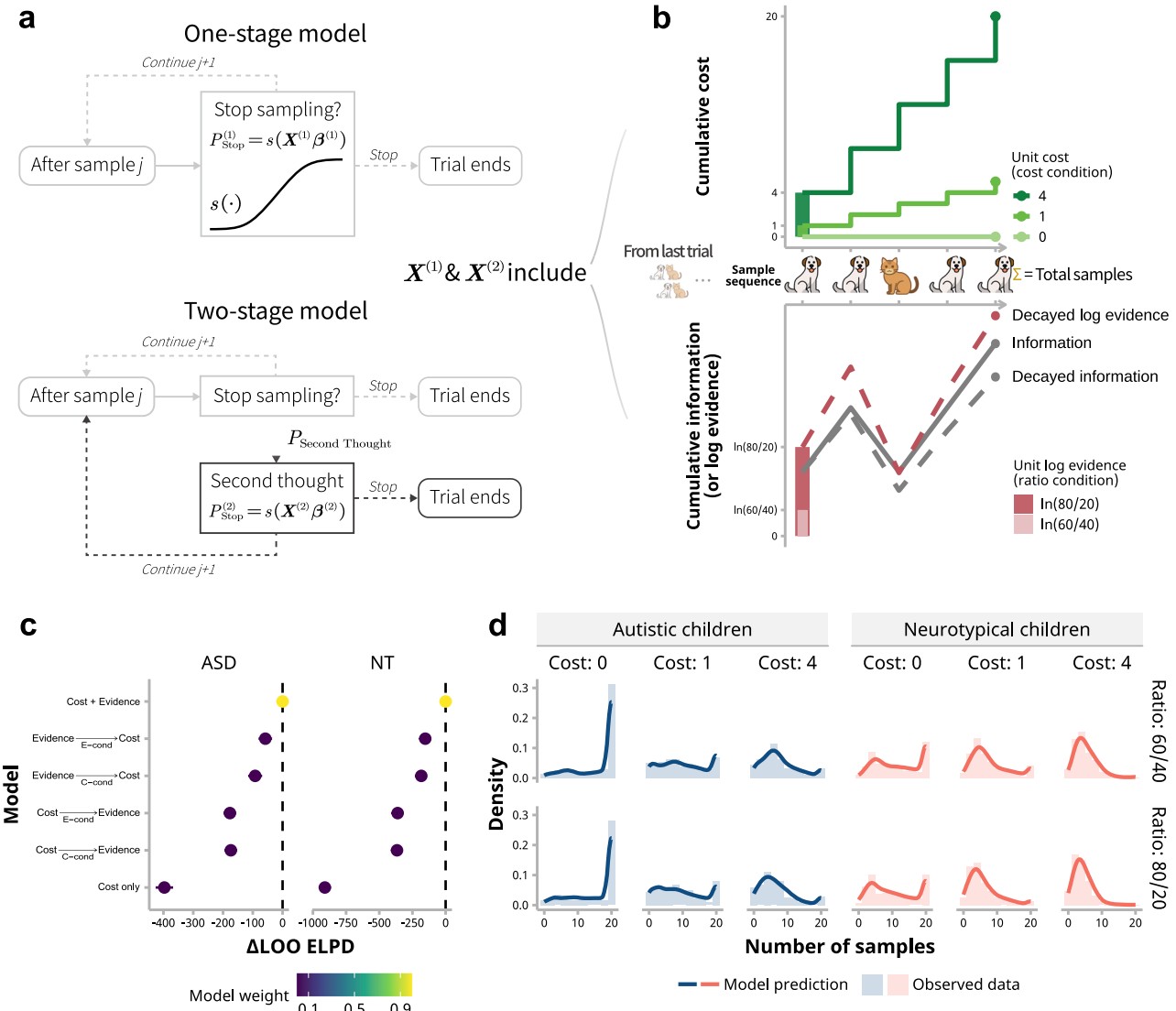

**Fig. 3 | Computational modeling of sampling choices in children. a** Model diagram. One-stage models (top) determine the probability of stopping sampling based on decision variables. Two-stage models (bottom) incorporate a potential second stage where the decision to stop sampling is reassessed, influenced by a different set of decision variables. **b** Decision variables categorized into cost-related and evidence-related groups. Cost-related variables include unit cost, total samples, and cumulative cost, whereas evidence-related variables consist of unit log evidence, total decayed information, cumulative decayed log evidence, along with the sample number and correctness of the last trial. **c** Model comparisons. LOO ELPD (leave-one-out expected log predictive density) signifies the model evidence, with higher values indicating a better fit to the data. ΔLOO ELPD represents the difference in LOO ELPD relative to the best model (i.e., Cost + Evidence), where a value nearer to zero means being closer to the best model. Error bars denote standard errors. Colors reflect model weights; heavier weights suggest a better model fit. **d** The best model's predictions and the actual data shown as the distributions of sample numbers. Solid lines depict model predictions and histograms indicate observed data ($n = 31$ autistic children, $n = 41$ neurotypical children).

LMMs were estimated using "afex" package[35], with degree of freedom approximated using Satterthwaite method to obtain $p$ values[36]. Specifications of random effects followed parsimonious modeling[37]. For significant fixed effects, "emmeans" package was used for post hoc contrasts[38], and the effect sizes and confidence intervals were estimated from $F$ statistics or $t$ statistics with approximated degree of freedom. Statistical multiplicity was controlled by a single-step adjustment using multivariate $t$ distributions[39].

For correlation analyses we conducted in the manuscript, including those between autistic traits, behavioral measures, and computational model parameters, we used Spearman correlations. To guard against inflated Type I error due to multiple testing, we used the false discovery rate and reported adjusted $p$-values by default unless stated otherwise.

## Behavioral modeling

***Model formulation***. To account for the difference in the cognitive mechanism behind the sampling decision of children, we adopted six computational models described in Lu et al.[22], which were two one-stage models and four two-stage models (Fig. 3a; also Supplementary Table 2 for model formulation and specifications). The first-stage models described what information children integrated to achieve the sampling decision. The second-stage models further distinguished how they made the decision in potentially two steps, with different information considered in each step. The information that the children were assumed to integrate could be categorized into cost-related (e.g., cumulative costs during sampling) and evidence-related decision variables (e.g., cumulative log evidence; Fig. 3b). Specifically, in one-stage models, the probability of stopping sampling at either stage on the $i$-th trial after having drawn $j$ samples is as a function of

the linear combination of $K$ decision variables (DVs) via a logistic function:

$$p_{ij} = \frac{1}{1 + e^{-X_{ij}}}, \qquad (2)$$

$$X_{ij} = \sum_{k=1}^{K} \beta_k DV_{ijk}. \qquad (3)$$

Four one-stage models differed in the combination of decision variables:

1. Cost-only model (denoted Cost only): cost-related variables, including unit cost per animal sample (categorical: 0, 1, 4; 0 was set as the reference level), number of animals sampled, and total sampling cost (product of the former two terms),
2. Cost and evidence with decay model (denoted Cost + Evidence): both cost-related and decayed evidence-related variables. On top of Cost only model, Cost + Evidence further includes unit log evidence per animal sample (i.e., $\ln\frac{80}{20}$ or $\ln\frac{60}{40}$), absolute value of cumulative information (cumulative information refers to the difference between the numbers of "cat" and "dog" samples; e.g., it would be one when there show 3 cats and 2 dogs), total log evidence (product of the former two terms), and the correctness and the number of animal samples in the last trial.

In models with decayed evidence, cumulative information (CI) is modulated by a decay parameter $\alpha$:

$$CI_{ij} = \begin{cases} 0 & j = 0; \\ \alpha CI_{i,j-1} + 1 & j > 0, \text{ after seeing a dog;} \\ \alpha CI_{i,j-1} - 1 & j > 0, \text{ after seeing a cat.} \end{cases} \qquad (4)$$

A value of the decay parameter near 0 indicates a tendency to consider only the most recently obtained information, whereas a value closer to 1 would mean considering all evidence with nearly equal weight.

In two-stage models, sampling choices may involve two decision stages, with the probability of reaching the decision of stopping sampling in each stage being

$$p_{ij}^{\text{Stage 1}} = \frac{1}{1 + e^{-X_{ij}^{\text{Stage 1}}}} \qquad (5)$$

$$p_{ij}^{\text{Stage 2}} = \frac{1}{1 + e^{-X_{ij}^{\text{Stage 2}}}} \qquad (6)$$

The overall probability of stopping sampling can be written as

$$p_{ij} = p_{ij}^{\text{Stage 1}} + \left(1 - p_{ij}^{\text{Stage 1}}\right) p_{ij}^{\text{sec}} p_{ij}^{\text{Stage 2}} \qquad (7)$$

Four second-stage models differed in both the combination of decision variables and the second-thought probability, including:

3. Cost $\xrightarrow{\text{C−cond}}$ Evidence: the first stage and the second-thought probability were both determined by cost-related variables, whereas the second stage was determined by evidence-related variables,

4. Cost $\xrightarrow{\text{E−cond}}$ Evidence: the first stage was determined by cost-related variables, but the second-thought probability was controlled by evidence conditions,

5. Evidence $\xrightarrow{\text{C−cond}}$ Cost: the first stage was determined by evidence-related variables, but the second-thought probability was controlled by cost conditions,

6. Evidence $\xrightarrow{\text{E−cond}}$ Cost: the first stage and the second-thought probability were determined by evidence-related variables.

When the second-thought probabilities were conditional on cost conditions, $p_{ij}^{\text{sec}} = p^{\text{zero cost}}$, $p^{\text{low cost}}$, and $p^{\text{high cost}}$ respectively for the zero-, low-, and high-cost conditions; when conditional on evidence conditions, $p_{ij}^{\text{sec}} = p^{\text{low evidence}}$ and $p^{\text{high evidence}}$ respectively for the low- and high-evidence conditions. In general, model parameters to be estimated included $\beta_k$ that represents the effects of decision variables, decay parameter $\alpha_{\text{Decay}}$, and second-thought probabilities $p_{ij}^{\text{sec}}$ if two-stage models (Supplementary Table 2).

**Model fitting and comparison.** We fitted models separately to the behavioral data of neurotypical and autistic children, using hierarchical Bayesian estimation with Hamiltonian Monte Carlo implemented in Stan and cmdstanr package in R[40,41]. In contrast to Lu et al. using maximum likelihood estimation, the hierarchical Bayesian estimation allows incorporation of prior information and sharing information across participants, which is especially beneficial for relatively small sample sizes and provide more informative and robust results.

Each of four separate Markov chains with randomized initial values took an adaptive warm-up with 5000 samples from the posterior to prevent dependence on the initial values, and then each chain took another 10,000 samples. This resulted in 40,000 samples from the posterior for each parameter. For complex hierarchical models, in accordance with the standard practice in Stan, parameters were sampled using non-centered parameterization; that is, parameters were independently sampled from a standard normal distribution before transformed to an appropriate range. Convergence between chains was confirmed based on Gelman-Rubin $\hat{R}$ of all parameters being less than 1.01 and no systematic divergent transitions. All individual-level parameters were assumed to be sampled from the corresponding group-level normal distributions, whose mean and standard deviation were estimated from the data (parameter specification of hierarchical models see Supplementary Methods 4).

Model comparison was performed based on the pointwise out-of-sample prediction accuracy of the model, which was evaluated by the Bayesian leave-one-out cross validation estimate of expected log predictive density (LOO ELPD) and implemented in "loo" package in R[42,43]. Moreover, models could be compared in terms of the model weights from pseudo-Bayesian model averaging with Bayesian bootstrapping (Pseudo-BMA+)[44]. The model weights were calculated taking into account the penalty for model complexity. Models with higher ELPD or Pseudo-BMA+ model wights suggest better performance while having a parsimonious model specification. All models were compared within each group of children (i.e., in autistic and neurotypical children separately). To better illustrate their influence, we conducted simulations based on the best-fitting model and by varying parameters of interests one by one within the range estimated from the children's data while keeping other parameters fixed. We applied the exact stimulus sequences that each child encountered during the experiment for the simulation, rather than creating hypothetical scenarios, so that the simulations are grounded in the data and allow us to systematically understand how each computational mechanism translates into observable behavior.

**Simulation-based calibration.** To verify our modeling and inference processes, we conducted simulation-based calibration (SBC) analysis. SBC can validate both model specification and inferential algorithms by examining discrepancies between the prior distribution and the data-averaged posterior[45]. The procedure draws $N$ parameter samples from the prior distribution and simulates $N$ datasets using the generative model. The model is then fitted to each simulated dataset to obtain posterior distributions, from which $D$ samples are drawn. We calculate the rank of each simulated parameter value within its corresponding $D$ posterior samples. Under correct model specification and inference, the ranks of

the parameters of interest should follow a uniform distribution over [0, D], which can be visually inspected.

We simulated $N = 1000$ datasets using our target model (the best-fitting model), the stimulus sequence from the autistic children group, and parameters sampled from the modeling priors. We obtained $D = 400$ post-warmup draws across two chains after thinned by 10 to reduce autocorrelation (1000 warmup draws discarded per chain). The ranks of all parameters of interest distributed uniformly, and the posterior means recovered the simulated "true" values effectively. Additionally, the recovered simulated parameters showed strong correlations with the true simulated parameters, confirming high recoverability for our model estimates even when including participants with fewer trials (see Supplementary Methods 5). All SBC analyses were conducted using the R package SBC[46].

### Reporting summary
Further information on research design is available in the Nature Portfolio Reporting Summary linked to this article.

## Results
We recruited 32 autistic children and 41 age- and IQ-matched neurotypical children aged five to eight years old (Table 1; see also "Methods"). Children completed a gamified version of the bead task where they encountered dogs and cats from an unknown island and needed to determine which of two islands they were on based on the animals they had seen (Fig. 1). In each of 96 trials, children could sample up to 20 animals sequentially by pressing a button, deciding when to stop sampling and make their judgment. Children received 100 credits for correct responses, but with a deduction for each sample taken. Their goal was to earn as many credits as possible. The experiment contained six blocks combining three cost conditions (zero-cost, low-cost, and high-cost: 0, 1, or 4 credits per sample) with two evidence strength conditions (low-evidence: 60%:40% vs. 40%:60% dog-to-cat ratios for two islands, and high-evidence: 80%:20% vs. 20%:80%). With the varying experimental conditions, children would have to adjust their sampling strategies for maximal rewards, which would be directly reflected in their number of samples and the variation of that. All estimated marginal means of each group under each condition were summarized in Supplementary Table 3.

### Autistic children differed from neurotypical children in sampling strategy rather than in correct judgment
On average, autistic children earned significantly fewer credits in each trial than neurotypical children (LMM1; $M_{ASD} = 70.0$, $M_{NT} = 74.9$, $t(70.2) = -3.53$, $p = 0.002$, $d = -0.15$, 95% CI [−0.24, −0.06]), particularly in the low-cost ($M_{ASD} = 71.8$, $M_{NT} = 76.9$, $t(72.2) = -2.52$, $p = 0.04$, $d = -0.15$, 95% CI [−0.30, −0.005]), the high-cost ($M_{ASD} = 54.0$, $M_{NT} = 64.7$, $t(72.6) = -4.10$, $p < 0.001$, $d = -0.32$, 95% CI [−0.51, −0.13]), and the high-evidence condition ($M_{ASD} = 77.4$, $M_{NT} = 85.7$, $t(71.3) = -4.99$, $p < 0.001$, $d = -0.25$, 95% CI [−0.37, −0.14]). There was no significant difference between the two groups in the proportion of correct judgment (LMM2; $\chi^2(1) = 1.70$, $p = 0.19$). This implied that the main behavioral difference was pertinent to their sampling strategy rather than judgment (see Supplementary Note 1 for detailed analyses).

### Sampling behaviors: autistic children had significantly lower efficiency than neurotypical children
An effective sampling strategy to reach optimal decisions was to balance costs and information gain when sampling (see Fig. 1b). Sampling efficiency, measured as the expected gain per trial divided by maximum expected gain, is a better metric than gain credits since it takes condition differences into account. We found a significant interaction between sampling cost and evidence (Fig. 2b; LMM3: $F(2, 72.10) = 24.63$, $p < 0.001$, $\eta_p^2 = 0.41$, 95% CI [0.23, 0.54]). Children had higher efficiency in high-evidence conditions than low-evidence conditions when sampling was costless ($t(70.7) = 6.42$, $p < .001$, $d = 0.46$, 95% CI [0.29, 0.64]). In low-cost trials, performance was

better in low-evidence conditions ($t(72.2) = -2.87$, $p = 0.016$, $d = 0.19$, 95% CI [0.07, 0.44]).

The overall sampling efficiency of autistic children was significantly lower than NT children ($M_{ASD} = 90.9\%$, $M_{NT} = 93.5\%$; $F(1, 71.13) = 7.42$, $p = 0.008$, $\eta_p^2 = 0.09$, 95% CI [0.01, 0.24]). A significant Group × Cost interaction ($F(2, 71.24) = 10.23$, $p < 0.001$, $\eta_p^2 = 0.22$, 95% CI [0.07, 0.37]) further showed that autistic children had better performance compared with neurotypical children when there was no sampling cost (Fig. 2b; in zero-cost trials: $M_{ASD} = 96.2\%$, $M_{NT} = 93.0\%$, $t(70.7) = 2.83$, $p = 0.018$, $d = 0.43$, 95% CI [0.06, 0.80]). However, their sampling efficiency dropped down more with higher cost (in high-cost trials: $M_{ASD} = 82.1\%$, $M_{NT} = 91.6\%$, $t(71.0) = -3.53$, $p = 0.002$, $d = -1.30$, 95% CI [−2.19, −0.40]). We also found a significant interaction between groups and evidence levels ($F(1, 71.05) = 8.53$, $p = 0.005$, $\eta_p^2 = 0.11$, 95% CI [0.01, 0.26]) that autistic children were less efficient than NT children in high-evidence conditions (high-evidence: $t(71.2) = -3.26$, $p = 0.003$, $d = -0.48$, 95% CI [−0.81, −0.16]; low-evidence: $t(71.2) = -1.77$, $p = 0.12$, $d = -0.22$, 95% CI [−0.49, 0.05]; also see Supplementary Table 3).

### Autistic children's lower sampling efficiency came from their higher sampling variation
Our analysis of sampling efficiency showed that the group difference was modulated by sampling cost: the higher the cost, the worse autistic children performed relative to NT children (Fig. 2b). While greater systematic biases (i.e., oversampling or undersampling) were potential factors, higher sampling variation might also lead to lower efficiency (see Fig. 2c). Or, both bias and variation might play their roles. To discern these possibilities, we dissected children's sampling behaviors into biases from optimality (Fig. 2d) and variation (Fig. 2e).

Children in both groups exhibited oversampling in high-cost trials and undersampling in zero-cost and low-cost trials, except that they showed no systematic bias in low-cost, high-evidence trials (LMM4; Cost × Ratio interaction: $F(2, 70.06) = 273.55$, $p < 0.001$, $\eta_p^2 = .89$, 95% CI [0.84, 0.92]; Fig. 2d and Supplement Table 3). The significant group difference suggested neurotypical children tended to sample fewer pieces of information than autistic children overall ($F(1, 71.00) = 17.03$, $p < 0.001$, $\eta_p^2 = 0.19$, 95% CI [0.06, 0.35]), and they exhibited significant undersampling compared with optimal sample numbers averaged across all conditions ($M = -2.72$, $t(70.5) = -5.24$, $p < 0.001$, $d = -0.80$, 95% CI [−1.10, −0.50]), whereas autistic children showed no significant deviation ($M = 0.523$, $t(71.4) = 0.89$, $p = 0.61$, $d = 0.15$, 95% CI [−0.19, 0.50]). Notably, in zero-cost conditions where sampling efficiency monotonically increases with acquired samples, autistic children's tendency to sample more information led to higher accuracy and efficiency, showing the strength of their strategy in this scenario (Supplementary Table 3).

We then looked at the group difference in sampling variation. The analysis revealed a significant main effect of group, with autistic children showing greater variation in the number of samples taken across trials (Fig. 2e; LMM5: $F(1, 71.18) = 12.78$, $p = .001$, $\eta_p^2 = 0.15$, 95% CI [0.03, 0.31]). There was also a significant interaction between group and sampling cost ($F(2, 89.13) = 4.05$, $p = 0.021$, $\eta_p^2 = 0.08$, 95% CI [0.001, 0.20]), with higher variation of the autistic group in low- and high-cost conditions than the NT group ($t(72.3) = 4.19$, $p < 0.001$, $d = 1.20$, 95% CI [0.50, 1.90] and $t(85.9) = 3.46$, $p = .002$, $d = 0.75$, 95% CI [0.22, 1.27], respectively). Considering both sampling bias and sampling variation, it appears that the less undersampling of autistic children in zero-cost conditions likely resulted in their higher efficiency compared with neurotypical children, whereas the high variation in sampling across trials in costly conditions also contributed to the lower sampling performance of autistic children.

### Computational modeling: autistic children were less influenced by dynamic and global information when making sampling decisions
To understand the cognitive mechanisms behind the observed differences in sampling decisions between autistic and neurotypical children,

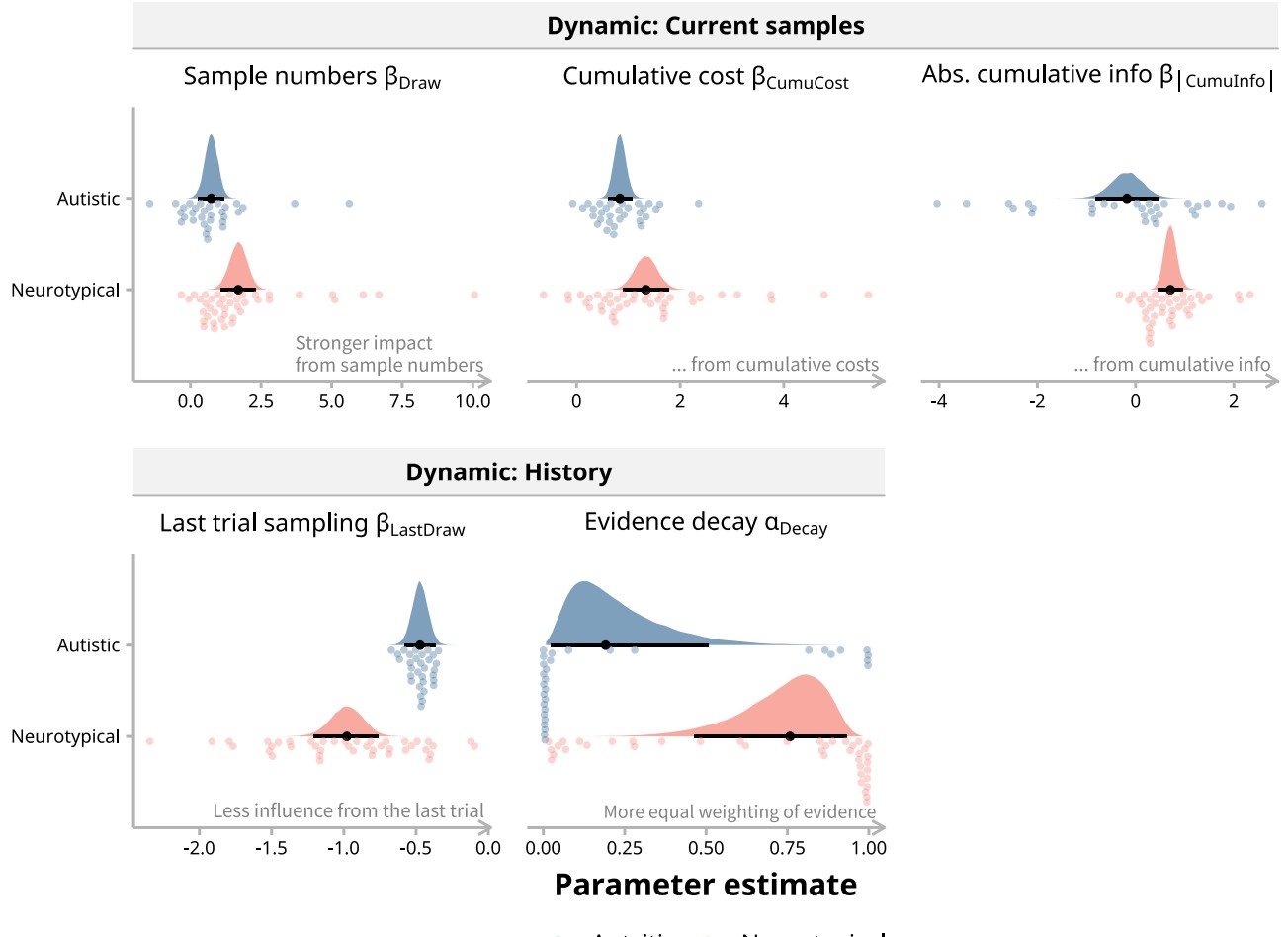

**Fig. 4 | Parameter estimates of dynamic variables from the best-fitting model.** The two groups of children showed significant differences in these parameters that gauge behavioral adaptability ($n = 31$ autistic children, $n = 41$ neurotypical children). The top row ("Dynamic: Current samples") captures the lower sensitivity of autistic children to changes as more samples accumulate within a trial: $\beta_{Draw}$, $\beta_{CumuCost}$, and $\beta_{|CumuInfo|}$ represent the influence of the sample number in the current trial, total costs accrued, and absolute value of accumulated information (with decay) during the current trial, respectively. The bottom row ("Dynamic: History") shows the influence from past information: $\beta_{LastDraw}$ represents the effect of the previous trial's sample number, and $\alpha_{Decay}$ determines how information gathered within a trial is weighted with lower values indicating greater weights on recent samples. These parameters reflect a flexible, adaptive process as they change with actions taken across and within trials. Data points denote individual parameter estimates, shaded areas represent group-level parameter distributions, and black markers with lines illustrate medians and 95% highest density intervals (HDIs) for these distributions.

we applied models from previous research using similar paradigm (Fig. 3a)[22]. These models examined how children integrated different types of information to make each sampling decision; that is, whether to take another sample after having samples collected was hypothesized to be a sigmoid function of multiple decision variables or a combination of multiple sigmoid functions (see Methods and Supplementary Table 2). We compared one-stage models, where children made sampling decisions based on a single evaluation of decision variables, with two-stage models, where children could potentially reassess their initial decision and decided again (Fig. 3a). The decision variables considered in these models fell into two categories: cost-related variables (e.g., unit cost, cumulative costs) and evidence-related variables (e.g., unit log evidence, accumulated evidence with decay; Fig. 3b). Each model provided a formal mathematical description of how these variables were combined to determine the probability of stopping sampling at any given point. We fitted these models to individual children's data within each group using hierarchical Bayesian methods, which allowed us to estimate both individual- and group-level parameters while accounting for the varying number of trials completed by each child. We then compared all the models within each group of children (i.e., in autistic and neurotypical children separately) to reveal by which model children's behaviors could

be better explained, based on the pointwise out-of-sample prediction accuracy as the model evidence.

Both groups showed that Model Cost + Evidence had the highest model evidence based on the number of samples (Fig. 3c), suggesting both groups of children shared a similar cognitive mechanism underlying information sampling. As the Cost + Evidence model indicated, children tended to combine cost-related and evidence-related decision variables all together to determine sampling choice in a single step. Based on the complete stimulus sequence during the experiment (including unpresented "cat" and "dog" stimuli that would have been shown if children had not stopped sampling), the posterior predictive checking further confirmed that Model Cost + Evidence predicted decision choices well for both groups (Fig. 3d).

To explore the specific computational processes that lead to the differences in behavioral outcomes between autistic and NT children, the population-level parameters of the winning Model Cost + Evidence were compared between the two groups (similar to Ahn et al.[47]; Fig. 4). Results showed that the two groups differed in the coefficients of the following decision variables (Fig. 4 and Supplementary Table 4): general stopping tendency in zero-cost trials ($\beta_0$), cost condition differences ($\beta_{H-0}$ for high-cost relative to zero-cost condition, and $\beta_{L-0}$ for low- to zero-cost difference), number of samples drawn in the current trial ($\beta_{Draw}$), cumulative sampling

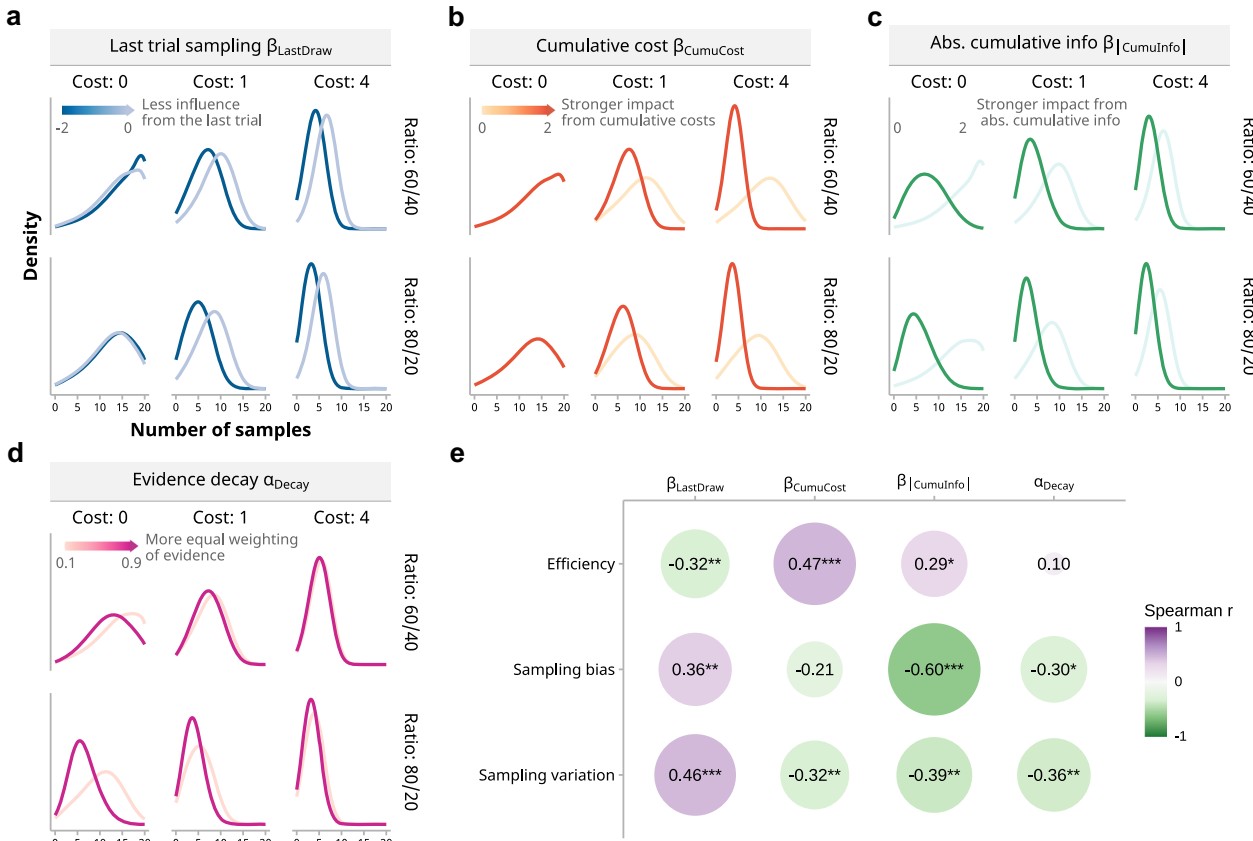

**Fig. 5 | Influence of dynamic decision parameters on sampling behavior.**
**a–d** Simulations illustrating the effect of dynamic parameters. The influence of $\beta_{\text{LastDraw}}$, $\beta_{\text{CumuCost}}$, and $\beta_{|\text{CumuInfo}|}$ was evident: a stronger impact from the last trial's sampling, cumulative costs and information resulted in fewer samples and reduced variation. Simulations for $\alpha_{\text{Decay}}$ indicated that a more equal weighting of evidence led to a reduction in sample numbers, particularly under conditions with more evidence and no cost. **e** Spearman correlation coefficients reveal the relationships between dynamic parameters (individual-level estimates of $\beta_{\text{LastDraw}}$, $\beta_{\text{CumuCost}}$, $\beta_{|\text{CumuInfo}|}$, $\alpha_{\text{Decay}}$) and behavioral metrics (efficiency, sampling bias, and sampling variation). Except for $\alpha_{\text{Decay}}$, the other three parameters could all significantly predict sampling efficiency of children. Significance levels are marked by asterisks (*$p < 0.05$, ***$p < 0.001$), with $p$-values adjusted for false discovery rate ($n = 72$ children).

cost ($\beta_{\text{CumuCost}}$), absolute value of cumulative information ($\beta_{|\text{CumuInfo}|}$), number of samples drawn in the last trial ($\beta_{\text{LastDraw}}$), and in the *decay* term of accumulated evidence ($\alpha_{\text{Decay}}$). The rest of the model parameters did not show noticeable group difference (see Supplementary Table 4 for summary statistics of parameters and group differences).

In addition to the cost- and evidence-related taxonomy, these model parameters can also be classified as "static" and "dynamic", by the nature of their coupled decision variables. This classification offers another perspective to understand the cognitive process differences between groups. Static parameters include $\beta_0$, $\beta_{\text{L-0}}$ and $\beta_{\text{H-0}}$ since they parallel the fixed sampling tendency and the differences between experimental conditions, which do not vary along with the sampling behavior. In contrast, dynamic parameters, such as $\beta_{\text{Draw}}$, $\beta_{\text{CumuCost}}$, $\beta_{|\text{CumuInfo}|}$, $\beta_{\text{LastDraw}}$, and $\alpha_{\text{Decay}}$, influence sampling decisions dynamically with each sample and trial. Then, we further examine how these static and dynamic parameters affect the sampling behavior of autistic and NT children.

### Reduced adaptation to dynamic information led to increased sampling variation in autistic children

First, we focused on the effects of the static parameters on sampling decisions. The autistic group had a much lower $\beta_0$, confirming a reduced tendency of stopping sampling and thus more samples on average under the zero-cost conditions (number of samples: $M_{\text{ASD}} = 15.56$, $M_{\text{NT}} = 10.97$, $t(70.6) = 3.86$, $p < 0.001$, $d = 1.35$, 95% CI [0.51, 2.19]). For $\beta_{\text{H-0}}$ and $\beta_{\text{L-0}}$, the autistic children exhibited higher values for both parameters compared to NT children, indicating their heightened sensitivity to nominal cost.

Specifically, in costly conditions, they reduced their sampling more significantly (difference in number of samples: $M_H - M_0 = -8.37$, $t(71.2) = -9.63$, $p < .001$, $d = -2.46$, 95% CI [−3.10, −1.83]; $M_L - M_0 = -5.74$, $t(70.8) = -6.88$, $p < 0.001$, $d = -1.69$, 95% CI [−2.30, −1.08]; see Fig. 2a), in contrast to NT children who showed less pronounced reduction ($M_H - M_0 = -6.04$, $t(70.1) = -7.99$, $p < 0.001$, $d = -1.78$, 95% CI [−2.33, −1.23]; $M_L - M_0 = -4.02$, $t(69.8) = −5.54$, $p < 0.001$, $d = -1.18$, 95% CI [−1.71, −0.65]). However, these static parameters alone could not explain why the sampling variation differed between the groups, since they predominantly govern general sampling tendency within a given condition.

Therefore, we turned to the dynamic parameters: $\beta_{\text{Draw}}$, $\beta_{|\text{CumuInfo}|}$, $\beta_{\text{CumuCost}}$ are conceptually similar for their influence dynamically dependent on current samples, whereas $\beta_{\text{LastDraw}}$ and $\alpha_{\text{Decay}}$ influence sampling decisions based on the history of the previous trial or samples within a trial (Fig. 3b and 4). The autistic group had group-level estimates of $\beta_{\text{Draw}}$, $\beta_{|\text{CumuInfo}|}$, and $\beta_{\text{CumuCost}}$ lower and closer to zero than the neurotypical group (Supplementary Table 4). This indicates that the autistic children were less sensitive to accumulated costs and evidence with increasing samples, whereas the neurotypical children were more cautious about sampling as they proceeded, reducing the number of samples they took. For $\beta_{\text{LastDraw}}$, autistic children had values that were larger and closer to zero compared to neurotypical children. The $\beta_{\text{LastDraw}}$ value of the autistic group, which was closer to zero, means the previous trial's sample count has little to no impact on the current decision, allowing for greater variation in sample numbers from trial to trial, which is consistent with the simulation (Fig. 5a). Finally, in the case of $\alpha_{\text{Decay}}$, the value of the neurotypical children approaching one

suggests that they consider all pieces of evidence more uniformly when deciding to stop sampling and are less swayed by recent evidence, leading to a more consistent sampling, in contrast to the autistic children with much lower $\alpha_{Decay}$. This is more evident in zero-cost and high-evidence conditions, where sampling decisions are not limited by cost, and evidence is easier to accumulate in one direction (Fig. 5d).

Based on these qualitative findings from the simulations, we further conducted correlation analyses between these dynamic parameters and children's sampling behaviors (Fig. 5e). Among all the parameters that the two groups differed in, $\beta_{CumuCost}$, $\beta_{|CumuInfo|}$, and $\beta_{LastDraw}$ could significantly predict overall sampling efficiency. $\beta_{LastDraw}$ had a significant negative correlation with average efficiency throughout the experiment ($r_S = -0.32$, 95% CI [−0.52, −0.09], $p = 0.026$). Such a negative correlation would be even stronger in high−cost ($r_S = -0.38$, 95% CI [−0.57, −0.16], $p = .002$) and high−evidence conditions ($r_S = -0.41$, 95% CI [−0.59, −0.19], $p = 0.003$). The direction of these correlations was consistent with the group differences in sampling efficiency and model parameters. $\beta_{CumuCost}$ and $\beta_{|CumuInfo|}$ were positively correlated with the efficiency ($\beta_{CumuCost}$: $r_S = 0.47$, 95% CI [0.24, 0.64], $p < 0.001$; $\beta_{|CumuInfo|}$: $r_S = 0.29$, 95% CI [0.06, 0.50], $p = 0.033$), which also mirrored the group difference in both the parameter and overall efficiency. We particularly investigated the effect $\alpha_{Decay}$ on sampling behavior since Bayesian theories of ASD predict strong reliance on incoming stimuli and weaker influence of past experience in autistic people. Although $\alpha_{Decay}$ was not correlated with the overall efficiency, as our simulation analyses predicted, it was negatively correlated with the efficiency under the zero-cost conditions ($r_S = -0.32$, 95% CI [−0.55, −0.11], $p = 0.021$; see Fig. 5d left panels). The parameter correlations with average sampling bias and variation were also consistent with the group differences (Fig. 5e; Supplementary Note 2).

These correlations, combined with the simulations, suggested that autistic children were less flexible and adaptive to the dynamic changes in the environment but more sensitive to the static and local nature of conditions. Such cognitive characteristics of autistic children further lead to their higher variation in sampling decisions, and ultimately relatively poorer sampling efficiency.

### Exploratory analyses: key behavioral findings with dimensional approach

We conducted exploratory dimensional analyses examining relationships between autistic trait scores (AQ-Child) and our key outcomes in a subsample of 51 children. These analyses replicated our group-based behavioral findings: children with higher autistic trait scores, particularly those scoring higher on the Social Skills subscale, showed lower efficiency in high-cost trials, increased overall sampling, and greater sampling variability (Supplementary Note 3). Individual-level computational parameter associations with AQ scores showed partial convergence with group-based findings, with only the evidence accumulation decay parameter $\alpha_{Decay}$ surviving multiple comparison correction (with AQ Social Skills scores: $r_S = -0.38$, 95% CI [−0.60, −0.11], $p = 0.031$; see Supplementary Note 3 for full details).

Within the autistic sample ($N = 17$ with ADOS-RRB scores available), we examined whether restricted and repetitive behavior (RRB) symptoms were associated with sampling variability. There was marginally significant positive association between ADOS-RRB scores and sampling variability ($\beta = 0.547$, $F(1, 16.59) = 4.29$, $p = 0.054$, $\eta_p^2 = 0.21$, 95% CI [0.00, 0.51]).

### Discussion

The study used a child-friendly bead task to investigate information sampling differences between autistic and neurotypical children. Both groups showed suboptimal sampling but for distinct reasons. Neurotypical children tended to undersample overall, particularly when sampling would not incur any cost, whereas autistic children showed higher trial-to-trial variability in costly conditions, resulting in lower sampling efficiency. Computational modeling further demonstrated that both groups considered cost- and evidence-related factors in their sampling decisions. However, the autistic group was less influenced by previous trials and cumulative costs and

information but favored recent evidence. Model simulations and correlational analyses confirmed that these factors contributed to the group differences in sampling decision efficiency, sampling bias, and variation.

Our block design experiment aimed to facilitate children's learning of the optimal policy, but autistic children did not exhibit stable sampling strategies within these blocks. Modeling results suggest this instability could result from autistic children's struggles to effectively use historical and dynamic information to adjust their strategies, coupled with heavier reliance on recent evidence in a trial. These findings are reminiscent of the Bayesian accounts of ASD, including the "hypo-prior" hypothesis[48] and the "high sensory precision" hypothesis[49–51]. According to this framework, autistic people place an imbalanced reliance on incoming sensory evidence over prior beliefs during perceptual inference[14,50,52], and have continuously high learning rates weakening the influence of past perception[13,51].

Our results resonate with the Weak Central Coherence account of ASD as well, which suggests a focus on details and local information, sometimes at the expense of broader and global context[53–55]. Our findings reflected such a detail-oriented cognitive style, where autistic children prioritized recent evidence (i.e., local information) over historical data from previous trials and samples (i.e., global information) in their decisions. Prior research on these theories predominantly addressed perceptual inference[56–65], but our study expands beyond perception in ASD to information sampling as a form of disambiguatory active inference[13,66]. Such actions (i.e., active inference) could link sensory processing differences to a broader range of autistic symptoms. Thus, our study underscores the importance of extending the Bayesian and Weak Central Coherence accounts of ASD beyond perception to include action, providing supporting evidence for understanding ASD as a difference in the balance between perception and action[13].

Our findings may also shed light on the mixed results found in previous studies on information sampling in ASD. Jänsch & Hare found that adults with Asperger syndrome sample less evidence compared to neurotypical adults in a bead task[19]. Similarly, Farmer et al. found that autistic adults sampled less information and made faster decisions, rather than adopting a deliberative sampling style[18]. However, also using a bead task, Brosnan et al. showed that autistic adolescents gathered more information compared to their neurotypical peers[20]. Vella et al. found that autistic participants collected more information to make more correct choices in a similar box task[21]. Although these inconsistencies likely reflect the substantial heterogeneity inherent in autism, with considerable variability in phenotypes, underlying neurotypes, and cognitive profiles across individuals, it is noteworthy that these studies provided limited insights beyond just how much information participants sampled. Additionally, methodological factors could contribute to these mixed findings. The absence of explicit sampling costs or internal cost measurements in these studies makes it difficult to assess the role of costs in sampling behavior.

As indicated by our results (Fig. 2d), whether autistic children oversample or undersample depends on the sampling cost structure. For instance, what might appear as "oversampling" in high-cost conditions reflects the same information-seeking tendency that proves advantageous when costs are low or none. Information sampling in autism may be characterized by different strategic priorities or styles rather than deficits per se. These results suggest that environmental demands differentially affect information sampling strategies between groups, consistent with broader literature on context effects and flexible behaviors in autism[22,67–74]. Therefore, focusing on different environmental structures and interactions with them may be more productive than stopping at general impressions of children's performance, although this interpretation warrants further investigation in larger samples.

Age differences also play an important role, with distinct profiles for adolescents in risk-taking[10,75–77] and information sampling[2,5]. These aspects might contribute to the discrepancies observed in previous studies[19,20]. Comparing our findings with Lu et al.[22], we noted similarities in greater sampling variability under costly conditions and lower efficiency but differences in underlying cognitive mechanisms. Autistic children had a cognitive process similar to neurotypical children but were more influenced by

static and local factors, whereas Lu et al. showed adults with more autistic traits adhered to a rigid and fixed strategy. Future research should systematically investigate how information sampling behavior of autistic individuals is regulated by contextual factors and how it develops. For instance, Crawley et al. demonstrated through computational models that individuals at different age stages, whether they had an ASD diagnosis, used different learning strategies in a probabilistic reversal learning task[68]. This suggests that learning mechanisms may become more complex as cognitive functions develop. Although direct comparisons between our study and Lu et al. could be challenging due to differences in diagnosis and experimental details, both studies collectively provide strong evidence that variations in information sampling behaviors may emerge early in autistic individuals and even persist across ages.

Our exploratory dimensional analyses provide additional context for understanding individual differences. The dimensional approach using AQ-Child scores successfully replicated the condition-specific efficiency patterns observed in group-based analyses, strengthening confidence in these findings. Computational parameter associations showed partial convergence with group-based findings, with only the evidence accumulation decay parameter surviving multiple comparison correction. Interestingly, the Social Skills subscale emerged as a particularly strong predictor in these analyses, suggesting that social comfort may relate to information sampling strategies even in non-social contexts. However, these results should be interpreted cautiously. The weaker statistical evidence for some parameter associations likely reflects the substantially smaller subsample size (particularly $N = 17$ autistic children with AQ data), measurement error inherent in retrospective parental reports, and the heterogeneity within autism discussed throughout this paper. Different subtypes may show distinct trait-behavior relationships that linear correlations cannot fully capture.

The lack of statistical power particularly necessitates cautious interpretation of our preliminary finding that higher restricted and repetitive behavior symptoms may be associated with greater sampling variability, though this pattern aligned with our main finding that trial-to-trial variation in sampling may reflect inflexible adaptation to contextual demands. Future research with adequately powered samples could examine whether different symptom profiles correspond to distinct cognitive strategy subtypes, potentially informing more personalized approaches to understanding decision-making in autism.

## Limitations

All the autistic children in our study had a formal ASD diagnosis by professional clinicians, but we enrolled the participants from different waves and sites so that children, particularly autistic children, underwent different assessments that served similar purposes. Nevertheless, this inconsistency prevents us from formally adopting a dimensional approach to correlate our modeling results with autistic symptom severity, as recently advocated in psychiatry[78,79].

A second limitation is the sample size (i.e., the number of participants). Our sample size was adequate for desired power based on our a priori power analysis, even using conservative effect size estimates (Supplementary Methods 2), leading to robust findings that accorded with those found in adults with varying autistic traits[22]. However, if we had resources to test more participants (verbal autistic children are relatively rare), we would be able to refine subtyping based on computational model parameters and offer more insights into individual and subgroup differences[80–82], given the heterogeneity at many levels in autism. It should also be acknowledged that the meta-analytic effect sizes can be biased by, such as selection or publication bias and the heterogeneity between studies across domains.

Future studies could benefit from more comprehensive assessments and larger and more diverse samples to achieve not only greater statistical power but also deeper insights into individual and subgroup differences.

## Conclusion

In summary, we show that autistic children perform differently in an information sampling task compared to neurotypical children of the same age and intelligence level. Using computational modeling, we find that autistic children are less affected by dynamic changes between and within trials and more inclined to weigh recent pieces of evidence, resulting in greater behavioral instability and lower sampling efficiency. The information sampling behaviors of autistic children resemble those of adults with more autistic traits, despite the strategic differences probably due to ages. These findings reveal developmental differences in a fundamental cognitive function at an early stage and may provide empirical support for and even expand the Bayesian framework of ASD.

## Data availability

All the behavior data supporting the analyses and conclusions of the article are available at https://doi.org/10.17605/OSF.IO/WDTQ2.

## Code availability

All the codes replicating the tables, figures, and statistical analyses of the article are available at https://doi.org/10.17605/OSF.IO/WDTQ2.

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

## Acknowledgements

This work was partly supported by the National Science and Technology Innovation 2030 Major Program (2022ZD0204800 to H.Z.), National Natural Science Foundation of China (32271116 and 32571244 to L.Y., and 32471152 to H.Z.), Clinical Medicine Plus X—Young Scholars Project of Peking University, the Fundamental Research Funds for the Central Universities (to L.Y., and Grant No. PKU2023LCXQ023 and PKU2024LCXQ046 to H.Z.), and funding from Peking-Tsinghua Center for Life Sciences. The funders had no role in study design, data collection and analysis, decision to publish or in the preparation of the manuscript. We would like to thank all the children and their parents for their participation. We are also thankful to Tianbi Li, Yixiao Hu, Zheng Wang, Xing Su, Luoyuan Zhang, Lu Chen, and Qingdao Elim School for their generous assistance with the study.

## Author contributions

H.L. collected, analyzed, and interpreted the data and drafted the manuscript. H.Z. and L.Y. provided supervision and funding acquisition. All authors participated in the conceptualization, reviewing, and editing of the manuscript.

## Competing interests

The authors declare no competing interests.
