## [Transparent Peer Review file · Communications Psychology]

Autistic children sample costly information with increased variability due to inflexible updating

Corresponding Author: Professor Hang Zhang

Version 0:

Decision Letter:

Dear Dr Zhang,

Thank you for your patience during the peer-review process. Your manuscript titled "Beyond over- or under-sampling: Autistic children's inflexibility in sampling costly information" has now been seen by 2 reviewers, whose comments are appended below. You will see that they find your work of some potential interest. However, they have raised quite substantial concerns that must be addressed. In light of these comments, we cannot accept the manuscript for publication, but would be interested in considering a revised version that fully addresses these serious concerns.

We hope you will find the Reviewers' comments useful as you decide how to proceed. Should additional work allow you to address these criticisms, we would be happy to look at a substantially revised manuscript. If you choose to take up this option, please highlight all changes in the manuscript text file, and provide a detailed point-by-point reply to the reviewers.

Editorially, we consider the issue of power critical for further consideration of a revised version. To this end, please perform a sensitivity analysis to provide evidence that the study is sufficiently powered at an alpha level of 80% or higher to detect the smallest theoretically or pragmatically meaningful effect size. Please note that we are not asking for a post-hoc power analysis for the effect size found in the study; the smallest effect size of interest requires a scientific justification on which we will seek reviewer feedback (e.g. Lakens D, Scheel AM, Isager PM. Equivalence Testing for Psychological Research: A Tutorial. *Advances in Methods and Practices in Psychological Science*. 2018;1(2):259-269. doi:10.1177/2515245918770963). If the sensitivity analysis confirms the Reviewer's concerns about sample size, we would require a replication in a sufficiently powered sample to consider a revision for further review. Please also address the concerns regarding the computational modelling expressed by Reviewer #2, including a parameter recovery.

I am attaching a checklist that details critical reporting requirements for the revised manuscript. Please attend to each item and ensure your manuscript is fully compliant. We are requesting that your manuscript aligns with these requirements as this facilitates the evaluation of your manuscript, reducing delays in re-review and potential future acceptance. If your revised manuscript is not aligned with these requests on major issues, such as those concerning statistics, it may be returned to you for further revisions without re-review. Additional information can be found in our style and formatting guide Communications Psychology formatting guide.

If the revision process takes significantly longer than five months, we will be happy to reconsider your paper at a later date, provided it still presents a significant contribution to the literature at that stage.

Please use the following link to submit your
- revised manuscript,
- point-by-point response to the referees' comments,
- cover letter (as a separate document),
- the Editorial Policy Checklist (see below),
- the Reporting Summary (see below), and
- the completed Editorial Request Table (attached):

Link Redacted

Thank you for the opportunity to review your work.

Best regards,

Saloni Krishnan

Saloni Krishnan, PhD
Editorial Board Member
Communications Psychology
orcid.org/0000-0002-6466-141X

REVIEWER EXPERTISE:

Reviewer #1: autism, decision making

Reviewer #2: computational modelling, decision making, development

REVIEWER REPORTS:

Reviewer #1 (Remarks to the Author):

This is an interesting study on an important topic to understand the debilitating intolerance of uncertainty that drives anxiety and contributes to feelings of social confusion in many autistic people. It's useful to start with children to watch the development with time. The task seems well-suited to the age. I have some major concerns and suggestions and a number of minor ones listed below. I like the ideas and analyses and would like to see the paper come to fruition but estimate (power analysis would be needed) at least double the sample size is needed to support the analyses.

- One pressing concern is the low sample size. There was no power analysis reported. It is noted that several other studies used a similar sample size but that doesn't make this one robust – continuing to make big assumptions from too little data just perpetuates bad science.
 - o Many of the statistical tests hovered around statistical significance but without adequate power it is impossible to interpret reliably. There is an imbalance in sample size between comparison groups which also affects the validity of the statistical tests. I believe this study is important but requires more statistical power to be meaningful.
 - o The authors state that it is difficult to recruit verbal autistic children – which is true. (do not use the term high-functioning however, which is too narrowly defined and now is passé). But difficulty finding an adequate sample isn't enough explanation for reporting on a sample that's simply too small, especially given the high number of tests and models being run (i.e., six models using just 24 autistic children)
- There needs to be a table of descriptive data for results. The statistical tests are reported but no means/variance results. Thus, for example, we don't even know the direction of effect for the significant tests.
- Within this context – it is really difficult to know which effects are reliable or not – the authors have overlooked the primary finding, as stated on Line 99: “Even though autistic children took fewer samples than neurotypical (NT) children 100 (Figure 2a; $F(1, 63.17) = 11.03, p < .001$), their overall earned credits did not significantly differ 101 (LMM1; $F(1, 63.32) = 3.90, p = .053$)”
 - o If I interpret this correctly, fewer trials overall but equal credits earned means that over the course of the whole experiment, the autism group was MORE EFFICIENT. I know there was a fancy formula used (Line 117) to show “the sampling efficiency of autistic children was marginally lower than NT children” but 1) marginal effects should not be claimed for support; it was not statistically significant – leave it at that; 2) this formula seems to be missing a key point: the autism group gained the same number of points in fewer guesses. That is the very definition of efficiency, is it not? Please explain if I am missing that.
- The difference between groups under high-cost trials is interesting. In essence: overall, autistic children are more efficient, but when the stakes are raised, they reduce their efficiency – a significant tradeoff that mirrors reports from everyday experience for autistic people. Things are ok when calm but stress rises easily and then decision making shuts down. I think this point can and should be argued, but only in the context of overall better efficiency. However, this all needs to be taken skeptically because of the low sample size.
- As I read further I see that any differences were because the neurotypical children were undersampling – this perhaps reflects a strength not a difficulty for autistic children; I think it supports a strengths-based approach rather than a deficits-

based approach for the autism group

- The computation modeling section (especially beginning line 177) is running many, many tests with a very low sample size. I think this is problematic. If you run enough tests you will find something just by chance; there is a real need for a power analysis here
- The simulations are interesting but at this point very far removed from the data. There are a lot of assumptions being made with little actual data.

Minor suggestions:

- Line 50: Repetitive behaviors could also reflect inefficient information sampling in reducing uncertainty due to the prolonged time spent on redundant details
 - o May be reconceptualized according to the Bayesian framework as: Repetitive behaviors used as a strategy for reducing uncertainty may contribute to inefficient information sampling due to the prolonged time spent on redundant details.
 - o I'll leave it to the authors to decide if this is a valid interpretation but fits the phenotype better in my opinion
- Review of inconsistencies in previous literature, and interpretation of these results, **MUST EXPLICITLY** acknowledge the heterogeneity of autism phenotypes (and corresponding genotypes and neurotypes) and differences in sampling; this is a highly likely explanation for many differences. In the introduction and the discussion.
- Line 61 . However, information sampling may change significantly from childhood to adulthood 2,5,9,10 62 , the information sampling behavior of autistic children remains unclear
 - o An unclear sentence. Could be fixed by adding as information sampling may change...

I sign all my reviews. ~Mikle South

Reviewer #2 (Remarks to the Author):

Thank you for the opportunity to review this manuscript. In this paper, the authors examine information sampling strategies in a sample of autistic children and an IQ-matched sample of neurotypical children. They find autistic children exhibit greater sampling variability relative to neurotypical children. Using computational models, they demonstrate these behavioral findings can be explained by a bias for more recently gathered evidence by autistic children relative to neurotypical children. Overall, the authors' design is appropriate to test their research questions. The use of mixed effect models and computational models are also appropriate for the analysis of trial-by-trial data, which is a benefit given the relatively small sample size. I have two key comments that I think would be important for the authors to address to reassure the reader of the robustness of their computational analyses, as well as some further suggestions to strengthen the evidence for their conclusions.

The explanation and methods used for the computational modelling are appropriate to test the authors' hypotheses. However, I could not see any information about the parameter recovery: Are the models able to estimate free parameters from simulated data using hard-coded values?

Relatedly, given that six autistic children only completed 48 trials (as opposed to 96), it would be important to demonstrate the parameters are recoverable from data with a limited number of trials. This point also has implications for the hierarchical fitting procedure used to estimate free parameters, as my understanding from the text is that data from all autistic participants was used when estimating group-level parameters regardless of the number of trials they completed. If the models cannot reliably estimate parameters from a limited number of trials, this may skew group-level estimates for the group of autistic children and subsequently individual-level parameter estimates.

To further strengthen the evidence for their conclusions, the authors should examine whether scores on the AQ-Child are associated with their key behavioural and computational outcome measures. Should this continuous analysis not replicate the findings of the group-based analyses, the authors should highlight this in the discussion and consider possible explanations as to why their findings were not reproduced when autistic traits were entered as a continuous predictor of their outcome variables, rather than a grouping variable.

I would be interested to know, within the sample of autistic children, whether specific symptoms restricted and repetitive behavior were associated with sampling variability. This analysis could be reported as exploratory and with the appropriate caveats of the small sample size.

The introduction and discussion are well written. The authors show good recognition of the limitations of this study and need for replication in a larger, more heterogeneous population.

EDITORIAL POLICIES

We ask that you ensure your manuscript complies with our editorial policies and reporting requirements.

To that end, we require revised manuscripts to be accompanied by two completed items: a reporting summary that collects

information on study design and procedure, and an editorial policy checklist that verifies compliance with all required editorial policies

- <https://www.nature.com/documents/nr-reporting-summary.zip>>Nature Research Reporting Summary
- <https://www.nature.com/documents/nr-editorial-policy-checklist.pdf>>Editorial Policy Checklist

All points on the policy checklist must be addressed. Your revised manuscript can only be sent back to the referees if these checklists are completed and uploaded with the revision.

Notes: If you have submitted a Stage 1 Registered Report, Review, Primer, Comment, or Perspective you do not need to submit these forms. If you have already submitted these forms, you may disregard this request.

** Visit Nature Research's author and referees' website at <http://www.nature.com/authors>>www.nature.com/authors for information about policies, services and author benefits**

Communications Psychology is committed to improving transparency in authorship. As part of our efforts in this direction, we are now requesting that all authors identified as 'corresponding author' create and link their Open Researcher and Contributor Identifier (ORCID) with their account on the Manuscript Tracking System prior to acceptance. ORCID helps the scientific community achieve unambiguous attribution of all scholarly contributions. You can create and link your ORCID from the home page of the Manuscript Tracking System by clicking on 'Modify my Springer Nature account' and following the instructions in the link below. Please also inform all co-authors that they can add their ORCIDs to their accounts and that they must do so prior to acceptance.
<https://www.springernature.com/gp/researchers/orcid/orcid-for-nature-research>

If you experience problems in linking your ORCID, please contact the <http://platformsupport.nature.com/>>Platform Support Helpdesk.

Version 1:

Decision Letter:

Dear Dr Zhang,

Your manuscript titled "Beyond over- or under-sampling: Autistic children's inflexibility in sampling costly information" has now been seen by our reviewers, whose comments appear below. In light of their advice I am delighted to say that we are happy, in principle, to publish a suitably revised version in Communications Psychology.

We therefore invite you to revise your paper one last time to address the remaining concerns of our reviewers and a list of editorial requests. At the same time we ask that you edit your manuscript to comply with our format requirements and to maximise the accessibility and therefore the impact of your work.

EDITORIAL REQUESTS:

SUBMISSION INFORMATION:

In order to accept your paper, we require the files listed here <https://www.nature.com/documents/commsj-file-checklist.pdf> .

OPEN ACCESS:

Communications Psychology is a fully open access journal. Articles are made freely accessible on publication. For further information about article processing charges, open access funding, and advice and support from Nature Research, please visit <https://www.nature.com/commpsychol/open-access>

* **DATA AVAILABILITY:**

All Communications Psychology manuscripts must include a section titled "Data Availability" at the end of the Methods section. More information on this policy, is available in the Editorial Requests Table and at <http://www.nature.com/authors/policies/data/data-availability-statements-data-citations.pdf>

Link Redacted

Best regards,

Troy Lui

Troy Lui, PhD
Associate Editor
Communications Psychology

Saloni Krishnan, PhD
Editorial Board Member
Communications Psychology
orcid.org/0000-0002-6466-141X

REVIEWERS' COMMENTS:

Reviewer #1 (Remarks to the Author):

Thank you for further attention to power concerns and interpretation. The additional sample size helps. Importantly, additional interpretation guides provide needed modestly for conclusions. Understanding the roots of difficult decision making for autistic children and adults is important and this study contributes to that. I have no further suggestions. ~Mikle South

Reviewer #2 (Remarks to the Author):

Thank you to the authors for their thorough reply to my comments. The manuscript is much improved and my concerns about the computational modelling analyses have been addressed.

EDITOR:

Editorially, we consider the issue of power critical for further consideration of a revised version. To this end, please perform a sensitivity analysis to provide evidence that the study is sufficiently powered at an alpha level of 80% or higher to detect the smallest theoretically or pragmatically meaningful effect size. Please note that we are not asking for a post-hoc power analysis for the effect size found in the study; the smallest effect size of interest requires a scientific justification on which we will seek reviewer feedback (e.g. Lakens D, Scheel AM, Isager PM. Equivalence Testing for Psychological Research: A Tutorial. *Advances in Methods and Practices in Psychological Science*. 2018;1(2):259-269. doi:10.1177/2515245918770963). If the sensitivity analysis confirms the Reviewer's concerns about sample size, we would require a replication in a sufficiently powered sample to consider a revision for further review. Please also address the concerns regarding the computational modelling expressed by Reviewer #2, including a parameter recovery.

We sincerely thank our editor and both reviewers for their valuable comments, suggestions, and concerns. In this newer version of the manuscript, we have made substantial changes regarding but not limited to:

- Sample size and power analysis (Editor and Reviewer #1)
- Parameter recovery (Reviewer #2)
- Interpretation of the results (Reviewer #1) and other concerns raised by both reviewers.

Per the editor's and reviewers' concerns and suggestions, we conducted a mini meta-analysis of closely related studies in autism research to determine the smallest effect size of interest (Lakens et al., 2018 & 2021; included studies and effect sizes in Supplementary Table 4). Given the estimated effect $r = .406$ (95% CI [.324, .483]), a priori power analysis indicated that a total sample of 44 (or 59) would provide 80% (or 90%) power to detect this effect at $\alpha = .05$. Our original sample already met this threshold. To further strengthen our findings, we successfully recruited additional participants, expanding our autistic sample by 33%. Our final sample of 73 participants (32 autistic, 41 neurotypical) provides 82% power even using the more conservative lower bound estimate ($r = .324$), and all main findings remain robust with this expanded dataset.

With our new data (33% more participants in the autistic group), the main behavioral and modeling results pertinent to our hypotheses remained unchanged: autistic children showed lower sampling efficiency under costly conditions, which resulted from their choice decisions being less influenced by dynamic changes in the environment.

We have also addressed Reviewer #2's concerns by conducting parameter recovery analyses using simulation-based calibration method for hierarchical Bayesian models. Visual inspection on the rank statistics and recovered estimates of the parameters of interest suggested the model was well calibrated and free from bias from fewer trials of some autistic children.

We believe our manuscript has significantly improved after addressing all the comments brought up by Editor and both reviewers.

Reviewer #1 (Remarks to the Author):

This is an interesting study on an important topic to understand the debilitating intolerance of uncertainty that drives anxiety and contributes to feelings of social confusion in many autistic people. It's useful to start with children to watch the development with time. The task seems well-suited to the age. I have some major concerns and suggestions and a number of minor ones listed below. I like the ideas and analyses and would like to see the paper come to fruition but estimate (power analysis would be needed) at least double the sample size is needed to support the analyses.

We are truly grateful for your review, comments, and suggestions. We understand the concerns you raised for the sample size and the statistical power of our analyses, which are extremely important for robust and rigorous research. Therefore, we have collected more data from verbal autistic children as much as we could and provided both a priori power analysis and sensitivity analysis. We have made substantial changes for other issues to improve the quality of our manuscript. Please see the changes and replies below.

• One pressing concern is the low sample size. There was no power analysis reported. It is noted that several other studies used a similar sample size but that doesn't make this one robust – continuing to make big assumptions from too little data just perpetuates bad science.

We appreciate this important concern and have conducted the comprehensive power analysis, including a priori and sensitivity power analysis requested by the Editor. For our power analyses, we followed the guide from Lakens and colleagues (2018 & 2021) and determined a smallest effect size of interest based on a mini meta-analysis to obtain the approximate mean and lower bound estimate of the effect in the literature. This incorporated 57 effect sizes from 23 related studies (Supplementary Table 4). Based on the mean estimate of the effect size ($r = .406$), a sample of 44 (two groups of children in total) would reach 80% statistical power at $\alpha = .05$, which our previous sample had already satisfied. With the newly added sample, we now could achieve 82% power even using the lower bound of the estimate ($r = .324$).

With our new sample of 32 autistic children, our main findings remained unchanged, suggesting the robustness of our experiment design and results. However, we still acknowledge the value of larger samples and have added this as a key limitation requiring future replication (Lines 384-395).

We have now added the power analysis to the Method section and the more detailed power analysis in Supplementary Methods 2, and also removed the justification based on related studies using a similar sample size (Lines 410-417).

o Many of the statistical tests hovered around statistical significance but without adequate power it is impossible to interpret reliably. There is an imbalance in sample size between comparison groups which also affects the validity of the statistical tests. I believe this study is important but requires more statistical power to be meaningful.

We appreciate the reviewer's attention to statistical rigor. Upon review, after we updated the sample size, only 1 test in our new exploratory analyses approached marginal significance (Line 283), which was not central to our main hypotheses, and we did not intend to overinterpret such results. Our current sample includes 32 autistic children, representing an increase from our initial submission, and we have updated all results accordingly.

Our current sample consists of 32 autistic children and 41 neurotypical children, resulting in a group ratio of 0.78, which is a modest imbalance well within acceptable limits for our statistical approaches. More importantly, our analytical strategy of using (generalized) linear mixed models and hierarchical Bayesian modeling were specifically chosen to handle unbalanced designs. Both are robust to unequal group sizes and actually gain efficiency by modeling individual-level variation while pooling information across participants.

o The authors state that it is difficult to recruit verbal autistic children – which is true. (do not use the term high-functioning however, which is too narrowly defined and now is passé). But difficulty finding an adequate sample isn't enough explanation for reporting on a sample that's simply too small, especially given the high number of tests and models being run (i.e., six models using just 24 autistic children)

We thank the reviewer for pointing out the use of language, and we have updated the wording (to “verbal autistic children”) in our manuscript accordingly (Line 388). We have also updated our sample to include 32 autistic children (rather than the 24 mentioned by the reviewer), and we have already applied appropriate multiple comparison adjustment when needed to address the concern about multiple testing (Lines 494-495).

• There needs to be a table of descriptive data for results. The statistical tests are reported but no means/variance results. Thus, for example, we don't even know the direction of effect for the significant tests.

We appreciate the reviewer's suggestion and have added a descriptive table by groups and conditions for all behavioral measures (see Supplementary Table 1). In the manuscript, we have reported the direction of effects and means or variance as well.

• Within this context – it is really difficult to know which effects are reliable or not – the authors have overlooked the primary finding, as stated on Line 99: “Even though autistic children took fewer samples than neurotypical (NT) children 100 (Figure 2a; $F(1, 63.17) = 11.03, p < .001$), their overall earned credits did not significantly differ 101 (LMM1; $F(1, 63.32) = 3.90, p = .053$)”

o If I interpret this correctly, fewer trials overall but equal credits earned means that over the course of the whole experiment, the autism group was MORE EFFICIENT. I know there was a fancy formula used (Line 117) to show “the sampling efficiency of autistic children was marginally lower than NT children” but 1) marginal effects should not be claimed for support; it was not statistically

significant – leave it at that; 2) this formula seems to be missing a key point: the autism group gained the same number of points in fewer guesses. That is the very definition of efficiency, is it not? Please explain if I am missing that.

We thank the reviewer for this thoughtful observation. While our initial submission showed marginal effects, in our current expanded sample, our results have changed from the initial submission regarding the main group effects: autistic children earned significantly fewer credits than neurotypical children ($M_{ASD} = 70.0$, $M_{NT} = 74.9$, $t(70.2) = -3.53$, $p = .002$) as well as the overall sampling efficiency ($M_{ASD} = 90.9\%$, $M_{NT} = 93.5\%$; $F(1, 71.13) = 7.42$, $p = .008$) (Lines 103-110). We have updated the manuscript accordingly and agree with the reviewer that examining condition-specific performance provides more meaningful insights than overall averages when the interaction effects were significant.

The reviewer's interpretation highlights an important distinction between two valid efficiency concepts:

- Resource efficiency (reviewer's approach): Given the same input (samples), how much output (credits) is achieved?
- Performance efficiency (our approach): Given maximum possible rewards per condition, what percentage of optimal performance is achieved?

Our research question centers on information sampling and decision-making optimality in children under different environmental constraints. While resource efficiency is intuitive, performance efficiency better captures strategic quality. The performance efficiency distinguishes between participants who achieve good outcomes through optimal strategies versus those who succeed through luck or suboptimal strategies that happen to work in specific instances.

Beyond definitional considerations, our efficiency formula provides additional benefits:

1. Raw credits cannot be meaningfully compared across conditions because maximum achievable rewards differ systematically: high-cost conditions are limited to a maximum of ~50-85 points due to sampling costs; zero-cost conditions allow up to 100 points.
2. By using expected rewards, our measure focuses on decision-making quality rather than random outcome variation. In Fig 1b., there is large variation of bonus points between children with similar average number of samples, due to random outcomes; yet, they all fall around the expected points based on our formula.

This approach allows us to assess how different groups adapt their information sampling strategies to environmental demands, which is the core question in our theoretical framework. We have clarified this in the Methods section: "It measures approximation to optimal reward-maximizing behavior under different environmental demands." (Line 470)

• The difference between groups under high-cost trials is interesting. In essence: overall, autistic children are more efficient, but when the stakes are raised, they reduce their efficiency – a significant tradeoff that mirrors reports from everyday experience for autistic people. Things are ok when calm but stress rises easily and then decision making shuts down. I think this point can and should be

argued, but only in the context of overall better efficiency. However, this all needs to be taken skeptically because of the low sample size.

We appreciate this insightful interpretation, which was based on our original results. However, even though our overall efficiency findings have changed with the expanded sample, the observation about the Group x Condition (both cost and evidence conditions) interactions remains relevant (Lines 122-131). Therefore, we believed that the results would be more informative when focusing on specific conditions and between them, instead of stopping at general impression of children's performance.

The pattern suggests that environmental demands differentially affect information sampling strategies between groups, consistent with broader literature on context effects and flexible behaviors in autism (Baez & Ibanez, 2014; Crawley et al., 2020; Fujino et al., 2019; Geurts et al., 2009; Kourkoulou et al., 2013; Lawson et al., 2015; Lu et al., 2019; Van Eylen et al., 2011; Vermeulen, 2015). We have emphasized these interactions as a key finding warranting further investigation in larger samples in the discussion of limitations (Lines 335-340).

• As I read further I see that any differences were because the neurotypical children were undersampling – this perhaps reflects a strength not a difficulty for autistic children; I think it supports a strengths-based approach rather than a deficits-based approach for the autism group

We appreciate the reviewer's call for a strengths-based perspective, which we have adopted in our revision. The reviewer is correct that in the zero-cost conditions where the efficiency increases monotonically with the number of samples, autistic children sampled more than neurotypical children, representing a near-optimal strategy that led to higher efficiency. We have stressed it in the Results and reframed our Discussion to emphasize this finding (Lines 147-150, 335).

However, we also note that deviation from optimal sampling numbers alone do not fully account for efficiency differences. As shown in Figure 2c, even when the average number of samples approximates the optimum, higher trial-to-trial variability always reduces overall efficiency. This suggests that consistent application of sampling strategies within a given context, in addition to between-condition adaption, are key factors distinguishing the groups.

We have revised our Results section to present these patterns descriptively and our Discussion to interpret them through a neurodiversity-affirming lens, acknowledging both context-dependent strengths and difficulties and the multifaceted nature of strategic decision-making.

• The computation modeling section (especially beginning line 177) is running many, many tests with a very low sample size. I think this is problematic. If you run enough tests you will find something just by chance; there is a real need for a power analysis here

We thank the reviewer for raising this important concern about multiple comparisons. We have provided detailed power analysis and included more children in the autistic group for this revision. To guard against inflated Type 1 error due to multiplicity, for all frequentist tests (linear mixed models and correlation analyses), we have applied p-value correction methods such as false discovery rate and multivariate t-

value adjustment. We report corrected p-values throughout unless otherwise noted, which we have stressed in the methods of this revision (Lines 492-495). Beyond statistical adjustments, our findings show a coherent pattern across multiple converging analyses rather than isolated significant results, strengthening confidence in their reliability.

In the computational modeling section, we did not apply traditional multiple comparison corrections for model parameters because Bayesian hierarchical modeling that we used inherently guards against spurious effects through following mechanisms:

1. Regularizing priors: We used priors centered on zero for group-level model parameters, meaning the model assumes no group effects unless the data provide strong evidence otherwise.
2. Partial pooling: Extreme estimates are automatically pulled toward the group mean through hierarchical shrinkage, reducing false positive rates, especially for noisy data.
3. Joint posterior inference: Unlike frequentist approaches where each test has independent error rates requiring correction, Bayesian inference produces a single joint posterior distribution for all parameters simultaneously. Any comparison is simply a query of this already-computed distribution, not an independent test with its own error rate.

• The simulations are interesting but at this point very far removed from the data. There are a lot of assumptions being made with little actual data.

We thank the reviewer for this comment and appreciate the opportunity to clarify how our simulations are grounded in the data. When conducting the simulations, we used parameter values estimated from actual participant data for all parameters except the one being varied. Additionally, we applied the exact stimulus sequences that each child encountered during the experiment, rather than creating hypothetical scenarios. The parameter ranges we explored were also constrained to values observed in our actual sample, ensuring the simulations reflect realistic behavioral patterns.

We acknowledge that varying one parameter while holding others constant assumes these parameters can be manipulated independently, though in reality they may be somewhat correlated. However, this standard sensitivity analysis approach allows us to systematically understand how each computational mechanism translates into observable behavior. This helps interpret what abstract model parameters mean in concrete terms and demonstrates that our parameters capture meaningful cognitive processes rather than arbitrary curve-fitting.

We have clarified the data-grounded nature of our simulations in the revised manuscript to address this concern (Lines 578-584).

Minor suggestions:

• Line 50: Repetitive behaviors could also reflect inefficient information sampling in reducing uncertainty due to the prolonged time spent on redundant details
o May be reconceptualized according to the Bayesian framework as: Repetitive behaviors used as a strategy for reducing uncertainty may contribute to inefficient information sampling due to the prolonged time spent on redundant details.

o I'll leave it to the authors to decide if this is a valid interpretation but fits the phenotype better in my opinion

We thank the reviewer for the suggestion. We have revised the manuscript accordingly.

• Review of inconsistencies in previous literature, and interpretation of these results, MUST EXPLICITLY acknowledge the heterogeneity of autism phenotypes (and corresponding genotypes and neurotypes) and differences in sampling; this is a highly likely explanation for many differences. In the introduction and the discussion.

We appreciate the reviewer's suggestion and have explicitly acknowledged this in both the introduction and the discussion (Lines 58, 324-326).

• Line 61 . However, information sampling may change significantly from childhood to adulthood 2,5,9,10 62 , the information sampling behavior of autistic children remains unclear

o An unclear sentence. Could be fixed by adding as information sampling may change...

We thank the reviewer for pointing it out. We have corrected it in the revised manuscript now.

I sign all my reviews. ~Mikle South

Reviewer #2 (Remarks to the Author):

Thank you for the opportunity to review this manuscript. In this paper, the authors examine information sampling strategies in a sample of autistic children and an IQ-matched sample of neurotypical children. They find autistic children exhibit greater sampling variability relative to neurotypical children. Using computational models, they demonstrate these behavioral findings can be explained by a bias for more recently gathered evidence by autistic children relative to neurotypical children. Overall, the authors design is appropriate to test their research questions. The use of mixed effect models and computational models are also appropriate for the analysis of trial-by-trial data, which is a benefit given the relatively small sample size. I have two key comments that I think would be important for the authors to address to reassure the reader of the robustness of their computational analyses, as well as some further suggestions to strengthen the evidence for their conclusions.

We truly appreciate the feedback and suggestions from the reviewer. In this revision, we have included more analyses for parameter recoverability and some exploratory correlational analyses with autistic symptoms, which we believe have improved the quality of our manuscript. Please see our replies below.

The explanation and methods used for the computational modelling are

appropriate to test the authors' hypotheses. However, I could not see any information about the parameter recovery: Are the models able to estimate free parameters from simulated data using hard-coded values?

Relatedly, given that six autistic children only completed 48 trials (as opposed to 96), it would be important to demonstrate the parameters are recoverable from data with a limited number of trials. This point also has implications for the hierarchical fitting procedure used to estimate free parameters, as my understanding from the text is that data from all autistic participants was used when estimating group-level parameters regardless of the number of trials they completed. If the models cannot reliably estimate parameters from a limited number of trials, this may skew group-level estimates for the group of autistic children and subsequently individual-level parameter estimates.

We thank the reviewer for this important methodological question. For this revision, we have added a parameter recovery analysis for our winning model using simulation-based calibration in Methods section (Lines 585-603) and detailed analysis in Supplementary Methods 5. This method works by first simulating multiple sets of parameters from the priors, then generating synthetic behavioral data conditioned on those simulated parameters, and finally fitting the model to the independently simulated datasets. If the model specification or inference algorithms have any issues, the posterior intervals will not appropriately cover the true simulated parameter values. We can assess this by examining the distribution of the rank of each simulated parameter within its posterior draws. A uniform rank distribution indicates a well-calibrated posterior, while deviations from uniformity suggest potential problems with the inference process.

To directly address the reviewer's concern about the six autistic children who completed only 48 trials, we conducted our simulation-based calibration specifically for the autistic group. When simulating the behavioral data, we used the actual 48-trial stimulus sequences from these six children mixed with the regular 96-trial sequences from other participants, fully mimicking the model fitting procedure for the autistic group. This provides a realistic test of whether our hierarchical model can reliably recover parameters when data includes a mix of trial numbers.

Our results demonstrate that all parameters of interest were successfully recovered with well-calibrated posteriors, as evidenced by their rank statistics following uniform distributions (Supplementary Figure 3 for SBC results of group-level parameters of interest). Additionally, the recovered simulated parameters showed strong correlations with the true simulated parameters (all parameters, including individual-level estimates, Pearson correlations $r_s > .33$, $p_s < .001$, with almost all group-level parameters exceeding .97), confirming high recoverability for our model estimates even when including participants with fewer trials.

Figure R1. Simulation-based calibration results. Details are also given in the Supplementary Methods 5.

To further strengthen the evidence for their conclusions, the authors should examine whether scores on the AQ-Child are associated with their key behavioural and computational outcome measures. Should this continuous analysis not replicate the findings of the group-based analyses, the authors should highlight this in the discussion and consider possible explanations as to why their findings were not reproduced when autistic traits were entered as a continuous predictor of their outcome variables, rather than a grouping variable.

We thank the reviewer for this valuable suggestion. We have conducted dimensional analyses using AQ-Child scores as continuous predictors for a subsample of 51 participants (17 autistic, 34 neurotypical) who had AQ-Child data available.

The dimensional analyses successfully replicated our group-based behavioral findings. Children with higher autistic trait scores showed the same condition-specific efficiency pattern, with higher efficiency in zero-cost trials but lower efficiency in high-cost trials (AQ \times Cost interaction: $F(2, 49.01) = 3.57, p = .036$). This pattern was particularly pronounced for the Social Skills subscale (Social Skills \times Cost: $F(2, 44.99) = 5.18, p = .009$ in a model where all subscales were entered). Consistent with group analyses, higher AQ total/social skill scores predicted increased overall sampling ($\beta_{AQ} = 1.17, t(48.9) = 2.44, p = .018$; $\beta_{AQ-Social\ Skills} = 2.14, t(45) = 2.58, p = .013$) and greater sampling variability ($\beta_{AQ} = 0.09, t(49) = 0.64, p = .53$; $\beta_{AQ-Social\ Skills} = 0.59, t(45) = 2.34, p = .023$). Interestingly, the Social Skills subscale emerged as a particularly strong predictor across behavioral measures, suggesting that social (dis)comfort may relate to information sampling strategies even in non-social contexts.

Correlational analyses between individual-level parameter estimates and AQ scores showed patterns directionally consistent with group differences, but most did not survive multiple comparison correction. When examining the Social Skills subscale specifically, only the evidence accumulation decay parameter α_{Decay} —which also showed significant group differences—remained significant after FDR correction ($r_s = -0.38, p = .031$). Other parameters showed correlations in expected directions but did not reach corrected significance thresholds.

Several methodological factors may explain why behavioral results replicated more robustly than computational parameters. Our AQ-Child subsample ($N = 51$) is

substantially smaller than the full sample ($N = 73$), prominently for the autistic group ($N = 17$ out of 34), reducing statistical power particularly for detecting associations with individual model parameters. Additionally, AQ-Child as a retrospective parental report measure, which heavily relies on parents' memories on past events, may introduce measurement error that adds noise to correlational analyses. The substantial heterogeneity within autism that we discuss throughout the manuscript means our subsample may represent particular subtypes, or contain multiple subtypes with different trait-behavior relationships that simple linear correlations cannot fully capture. Finally, correlating point estimates from posterior distributions may be less sensitive than incorporating dimensional predictors directly into the hierarchical model fitting procedure.

The strong convergence for behavioral outcomes reinforces our main findings regarding condition-specific efficiency patterns and sampling variability. We have added the details of these dimensional analyses to the Supplementary Results 3, with a summary in the main text Exploratory Analyses and Discussion (Lines 270-279, 357-369).

I would be interested to know, within the sample of autistic children, whether specific symptoms restricted and repetitive behavior were associated with sampling variability. This analysis could be reported as exploratory and with the appropriate caveats of the small sample size.

We thank the reviewer for this interesting suggestion. We conducted exploratory analyses examining whether RRB scores were associated with sampling variability within the autistic sample. We emphasize that only 17 autistic children had ADOS-derived RRB measures available, substantially limiting statistical power and requiring cautious interpretation.

Using RRB scores as a continuous predictor of sampling, we found higher RRB was associated with greater sampling variability but failed to reach significance ($\beta = .547$, $F(1, 16.59) = 4.29$, $p = .054$). Although non-significant, this pattern directionally aligns with our main conclusion that inflexibility in adapting sampling strategies to contextual demands characterizes information sampling in autism. Higher RRB may reflect broader difficulties in flexibly adjusting behavior to environmental constraints, manifesting as greater trial-to-trial variability when cognitive demands increase. However, given the very small sample size and exploratory nature of this analysis, these findings require replication in larger samples before drawing firm conclusions. In the manuscript, we report this result conservatively given the limitations (Lines 280-283, 370-376).

The introduction and discussion are well written. The authors show good recognition of the limitations of this study and need for replication in a larger, more heterogenous population.

We are very grateful for the reviewer's recognition and thoughtful feedback throughout the review process. We believe the manuscript has been significantly strengthened by addressing all the concerns raised.

REVIEWERS' COMMENTS:

Reviewer #1 (Remarks to the Author):

Thank you for further attention to power concerns and interpretation. The additional sample size helps. Importantly, additional interpretation guides provide needed modestly for conclusions. Understanding the roots of difficult decision making for autistic children and adults is important and this study contributes to that. I have no further suggestions. ~Mikle South

We really appreciate Dr. South's suggestions and comments.

Reviewer #2 (Remarks to the Author):

Thank you to the authors for their thorough reply to my comments. The manuscript is much improved and my concerns about the computational modelling analyses have been addressed.

We are grateful for all the feedback and comments from Reviewer #2.